# Distinct effects of several ice production processes on thunderstorm electrification and lightning activity

Inès Vongpaseut<sup>1</sup> and Christelle Barthe<sup>1</sup>

<sup>1</sup>LAERO, Université de Toulouse, CNRS, UT3, IRD, Toulouse, France

**Correspondence:** Inès Vongpaseut (ines.vongpaseut@univ-tlse3.fr)

**Abstract.** Ice particles play a crucial role in shaping cloud electrification, affecting the intensity of lightning activity. Previous studies have found a change of electric activity with varying aerosols concentration or active secondary ice production processes (SIP). However, the electric response to those parameters can differ with different cloud conditions and interact between themselves. The Meso-NH model was used with the two-moment microphysics scheme LIMA coupled with an explicit electrical scheme. Three idealized storms with varying warm-phase thicknesses were simulated to examine their response to aerosol concentrations and SIP mechanisms.

Increasing the cloud condensation nuclei (CCN) or the ice nucleating particle (INP) concentration increases ice crystal concentration, non-inductive charging and lightning activity up to a threshold. The main ice production processes (heterogeneous, homogeneous nucleation or Hallett-Mossop mechanism) depend on the cloud base temperature, and the aerosol concentration. CCN concentration thresholds (1000-8000  $\rm cm^{-3}$ ) differ across all storms due to cloud base temperature, while the threshold for INP concentration is generally  $\sim 100 \rm \ L^{-1}$ . Higher CCN concentrations increase cloud water content, affecting charge polarity, but has a relatively limited impact on graupel mass.

SIP mechanisms significantly enhance non-inductive charging and lightning activity by increasing ice crystal concentrations, particularly at low altitudes where primary ice production is inactive. This promotes ice-graupel collisions and amplifies charge exchange in each grid cell. The intensity of SIP processes varies with the thickness of the warm-phase region. Raindrop shattering freezing is the most sensitive and requires a deep warm-phase, while Hallett-Mossop and collisional ice break-up produce abundant ice crystals in all storms.

#### 1 Introduction

Cloud electrification processes are a key element in understanding and anticipating the electrical structure of thunderstorms and their electrical activity. The non-inductive charging mechanism refers to charge separation during rebounding collisions between ice crystals and graupel in the presence of supercooled liquid water (Reynolds et al., 1957; Takahashi, 1978). It is known to be the leading process of deep convective cloud electrification (Norville et al., 1991; Helsdon Jr. et al., 2001). Although all laboratory studies agree on the dependence of the sign and amplitude of the separated charge on temperature and supercooled liquid water content (e.g., Takahashi, 1978; Jayaratne et al., 1983; Saunders and Brooks, 1992; Saunders and Peck, 1998), they can strongly differ in the position of the charge reversal line (see figure 1 in Takahashi et al. (2017) or figure 2 in

Emersic and Saunders (2020)). This charge reversal line represents the temperature and liquid water content conditions where the sign acquired by the hydrometeors reverses. These conflicting laboratory results may be the consequence of difficulties in reproducing natural conditions in a cloud chamber (Takahashi et al., 2017). According to the theory of relative diffusional growth rate (RDGR) (Baker et al., 1987), the temperature at which the exchanged electric charge reverses sign depends on the vapor diffusion growth rate of ice particles. The particle that grows faster by water vapor diffusion charges positively during collision. By examining the factors influencing the rate of vapor deposition growth of pristine ice and graupel, Emersic and Saunders (2020) concluded from their laboratory experiments that, among other factors, the size of ice crystals and the cloud supersaturation should be better characterized. Glassmeier et al. (2018) have performed calculations of the RDGR theory as post-processing of the COSMO model and explored its sensitivity to numerous parameters. They identified ice crystal size as the most important parameter of RDGR, followed by graupel size.

While ice crystals are a key element in cloud electrification, their formation is complex and can follow different pathways. Ice crystals can be formed via homogeneous freezing of supercooled droplets at temperatures below -35°C. At warmer temperatures, aerosols acting as ice nucleating particles (INP) are required to form ice crystals through heterogeneous nucleation. Consequently, aerosols are indirectly involved in cloud electrification via the formation of ice crystals and cloud droplets.

40

60

The influence of aerosols acting as cloud condensation nuclei (CCN) on thunderstorm electrification and subsequent lightning activity has been examined in various observation-based and numerical modeling studies. In general, observation-based studies indicate a correlation between increased total lightning or cloud-to-ground lightning activity and increased Aerosol Optical Depth (AOD) in different regions of the world (Shi et al., 2020; Proestakis et al., 2016; Dayeh et al., 2021; Wang et al., 2023; Altaratz et al., 2010). However, studies have observed a decline in lightning activity when the AOD exceeds a threshold value, which ranges from 0.25 to 1, depending on the study. The same behavior is obtained in modeling studies in which the total lightning activity increases with the CCN concentration (Mansell and Ziegler, 2013; Sun et al., 2021; Tan et al., 2017). All these studies argue that an increase in CCN concentration increases the concentration of cloud droplets while reducing their size. Collision-coalescence processes are thus diminished in favor of droplet transport in the mixed phase of the cloud. The release of additional latent heat leads to increased vertical velocities and ice crystal concentrations, which in turn promotes charge separation via the non-inductive process (van den Heever et al., 2006; Rosenfeld et al., 2008; Sun et al., 2021). Mansell and Ziegler (2013) also detected an optimal CCN concentration of approximately 2000 cm<sup>-3</sup>, at which total lightning activity is maximized. They attributed the sharp decline in lightning activity at very high CCN concentration to the reduced efficiency of the Hallett-Mossop ice multiplication process, as the size of cloud droplets becomes too small for effective rime splintering. As for the effect of INP concentration on cloud electrification, it has received limited investigation. Using a 1.5D aerosol-cloud bin model, Yang et al. (2020) showed that increasing INP concentration from 300 to 1300 L<sup>-1</sup> results in larger ice particles and enhanced charging rate. However, as shown by Fuchs et al. (2015) and Phillips and Patade (2022), the aerosol effect on cloud electrification is modulated by the temperature at the cloud base or by the warm-phase thickness of the cloud.

Secondary ice production (SIP) processes are recognized as major contributors to ice particle concentrations (e.g., Field et al., 2016; Korolev and Leisner, 2020). Several SIP processes have been identified from laboratory experiments and in situ measurements, and some of them are now parameterized in microphysical schemes enabling the study of their impact on the

development and microphysical structure of deep convective clouds (e.g., Sullivan et al., 2018; Qu et al., 2022; Grzegorczyk et al., 2025a). However, up to now, few studies have focused on their effect on cloud electrification. Yang et al. (2024) simulated a cold-season thunderstorm with four different SIP processes. They showed that the SIP processes are active at different times in the storm lifecycle and at different altitudes, and that the rime splintering mechanism produces the higher ice crystal concentration in this case study. Through an enhancement of graupel and ice crystal production, an increase in the low-level positive charging rate on graupel is simulated, and the modeled flash rate shows better agreement with observations. In a simulation of a cold-based continental thunderstorm, Phillips and Patade (2022) found that the most active SIP process was breakup during ice-ice collisions. This process, acting as a sink of liquid water content, has the ability to alter the polarity of the charge graupel acquires and, consequently, the electric charge structure. They also stated that the cold cloud base temperature makes SIP processes less sensitive to CCN concentration. Several studies have shown that cloud electrification is sensitive to aerosol concentration and SIP processes, but the impact of CCN, INP, and SIP processes on cloud electrification has been studied separately. Moreover, their impact can be modulated by the cloud base temperature and the warm-phase thickness, while most studies have focused on a single case study (Mansell and Ziegler, 2013; Tan et al., 2017; Yang et al., 2020; Sun et al., 2021; Yang et al., 2024; Huang et al., 2025).

Therefore, to assess the impact of ice production on cloud electrification and lightning activity, three idealized thunderstorms with different cloud base temperatures are simulated using the 3D cloud-resolving model Meso-NH (Lac et al., 2018) with a quasi two-moment microphysical scheme (Vié et al., 2016) coupled to an explicit electrical scheme (Barthe et al., 2012). The simultaneous variation of CCN and INP concentrations enables the evaluation of their combined contribution to ice production and their subsequent impact on cloud electrification. The study also tests the sensitivity to three SIP processes. This paper is organized as follows. The model set-up and the methodology are presented in Section 2. Section 3 presents the results on aerosol concentrations, while Section 4 discusses the impact of SIP processes. Section 5 provides a summary.

#### 2 Simulation framework

## 2.1 The Meso-NH model

In this study the 3D atmospheric mesoscale model Meso-NH (Lac et al., 2018) in version 5-7 is used to simulate different idealized thunderstorms. Meso-NH is the high-resolution limited area research model of the French community and enables performing simulations of idealized cases or real meteorological situations over complex terrain initialized and forced at the lateral boundaries from model outputs. The model has a complete set of physical parameterizations: convection, turbulence, microphysics, aerosols, chemistry, radiation, atmospheric electricity... In the present study, a focus is done on the explicit coupling between the quasi two-moment microphysical scheme LIMA (Vié et al., 2016) and the electrical scheme CELLS (Barthe et al., 2012).

### 2.1.1 Microphysical scheme




LIMA (Liquid Ice Multiple Aerosols, Vié et al., 2016) is a quasi two-moment microphysics scheme which includes five types of hydrometeors: cloud droplets, raindrops, pristine ice crystals, snow/aggregates, and graupel. It predicts the mass mixing ratio for all five categories of hydrometeors and the number concentrations only for cloud droplets, raindrops, and ice crystals. LIMA includes a representation of the aerosols as multi-lognormal distributions of aerosols acting as CCN or INP. Details on the LIMA microphysics scheme can be found in Vié et al. (2016).

Here we focus on the different ice crystal production processes: heterogeneous and homogeneous ice nucleation, and SIP processes. Heterogeneous ice nucleation is parameterized as proposed by Phillips et al. (2008, 2013). Activated INP are computed by integration of a reference activity spectrum which depends on supersaturation and temperature. Ice crystals can also form through homogeneous nucleation of cloud droplets at temperatures below -35°C, with the homogeneous cloud droplets freezing rate taken from Eadie (1971). Three SIP processes are available in LIMA. First, the Hallett-Mossop process (HM), often referred to as rime splintering (Hallett and Mossop, 1974) produces an ice splinter each time a graupel is rimed with 200 droplets having diameters between 12 and 25 µm (Beheng, 1987) as described in Vié et al. (2016). Then, the collisional ice break-up (CIBU) mechanism deals with the production of ice splinters during collisions between fragile snow/aggregate particles and large and dense graupel particles (Vardiman, 1978; Takahashi, 1975; Yano and Phillips, 2011). The implementation of this parameterization in LIMA is described by Hoarau et al. (2018). It depends on the impact velocity between the two particles, and enables varying the number of ice fragments, which can also be randomized. Finally, the raindrop shattering freezing (RDSF) process was recently implemented in LIMA. It deals with ice splinters production during raindrop freezing. The number of fragments depends on the raindrop diameter (Lawson et al., 2015) and the probability of shattering has a Gaussian temperature dependency centered at 258 K, as introduced in Sullivan et al. (2018) on the basis of droplet levitation experiments. A general form of the equation describing the RDSF process can be written:

$$\frac{\partial n_i}{\partial t} = \alpha n_i n_r \tag{1}$$

where  $n_i$  and  $n_r$  are the particle size distribution of cloud ice and raindrops, respectively. An expression for  $\alpha$  is:

$$\alpha = \mathcal{N}_{\text{RDSF}} V_r \frac{\pi}{4} D_r^2 \tag{2}$$

where  $V_r$  is the impact velocity of a raindrop of size  $D_r$  at the surface of the ice crystal.  $\mathcal{N}_{RDSF}$  is the number of ice fragments per raindrop freezing and is parameterized as:

$$\mathcal{N}_{\text{RDSF}} = p_{sh}(T)\chi D_r^4 \tag{3}$$

 $p_{sh}$  is the shattering probability depending on temperature (T). According to Lawson (2015),  $\chi$  is set to  $2.5 \times 10^{13} \ \mathrm{m}^{-4}$  and  $p_{sh}(T) = 0.2N(258\mathrm{K}, 5\mathrm{K})$  where  $N(258\mathrm{K}, 5\mathrm{K})$  is a normal distribution centered around 258 K and with a variance of 5 K. The maximum of the shattering probability was found to be 20 % based on laboratory experiments (Leisner et al., 2014).

#### 2.1.2 The electrical scheme






The Cloud ELectrification and Lightning Scheme (CELLS) (Barthe et al., 2012) is implemented in Meso-NH and can simulate the electrification of clouds and their lightning activity. The scheme computes the evolution of the bulk charge carried by each type of hydrometeors and also takes into account free ions. Several non-inductive charge separation parameterizations are available which are all associated with collisions between a rimmed particle (graupel or snow/aggregates) and an ice particle growing mostly by deposition (ice crystal or snow/aggregates) (Takahashi, 1978; Saunders et al., 1991; Saunders and Peck, 1998; Tsenova et al., 2013). Inductive charging resulting from collisions between graupel and droplets in a preexisting electric field is also considered following the approach of Ziegler et al. (1991). While the cloud electrification scheme described in Barthe et al. (2012) was built upon the ICE3 one-moment bulk microphysics scheme (Pinty and Jabouille, 1998), recent developments in Meso-NH now allow the cloud electrification scheme to be used with the LIMA two-moments microphysics scheme. The electric field is computed at each time step following the procedure described in Barthe and Pinty (2007b) and is updated after charge neutralization by each lightning flash. Lightning flashes are triggered when the electric field exceeds a height-dependent threshold (Marshall et al., 1995). Flashes are described first as a bidirectional phase (vertical extension) and secondly, the branches spread horizontally according to a fractal law (see Barthe et al., 2012, for a full description of the lightning flash scheme).

## 2.2 Case studies and numerical set-up

Different environmental conditions may impact ice crystals formation pathways, and consequently cloud electrification. Therefore, numerical simulations of three contrasted idealized thunderstorms were performed. Figure 1 depicts the different cloud phases (warm, mixed, and cold) during the early electrification of the three simulated thunderstorms. The first case (hereinafter called WARM) is a tropical maritime thunderstorm. It has the warmest cloud base temperature (23°C) and is associated with the deepest warm phase (4 km). The second case (hereinafter called MID-WARM) is a continental case with a slightly warm cloud base (16°C). It has a 2.5 km deep warm phase, is less vertically developed (up to 10 km), and is associated with a shallow cold phase of about 1 km high. The last case (hereinafter called COLD) is a continental storm with a cold cloud base (10°C) and a very shallow warm phase (1.5 km).

All the simulations were performed with the same horizontal ( $\Delta x = \Delta y = 500 \text{ m}$ ) and vertical ( $\Delta z = 250 \text{ m}$ ) grid spacing. The WARM case is the thunderstorm observed on the  $18^{th}$  of July 2002 during the CRYSTAL-FACE (Cirrus Regional Study of Tropical Anvils and Cirrus Layers - Florida Area Cirrus Experiment; Leroy et al., 2009) experiment in southern Florida. Meso-NH was run for 1 hour with a 2.5 s time step over a domain of  $256 \times 256 \times 72$  gridpoints. A warm bubble of  $1.5^{\circ}$ C was used to trigger convection. The MID-WARM case was simulated using the sounding from Klemp and Wilhelmson (1978) (see Fig. S1 in supplement). A warm bubble of  $1.5^{\circ}$ C was also used to trigger convection. The simulation lasted 1 hour and was run with a time step of 3 s. A  $200 \times 200 \times 60$  gridpoints domain was used. The COLD case study is the 10 July 1996 thunderstorm observed during the STERAO (Stratospheric-Tropospheric Experiment: Radiation, Aerosols, and Ozone) experiment near the Wyoming-Nebraska-Colorado border. The initial sounding comes from Skamarock et al. (2000) (see Fig. S1 in supplement)

and is applied on a domain of  $270 \times 270 \times 72$  gridpoints. Unlike Skamarock et al. (2000), who used three warm bubbles to initiate this storm, here a single warm bubble was used. Indeed, the objective was not to reproduce the observed thunderstorm but to simulate a single cell storm with a cold cloud base to simplify the analysis. For each simulation, output files are available every 5 min. Microphysical budgets are calculated and integrated every 5 min over a 5 min period.

The aerosol populations acting as CCN and INP are prescribed using a single mode for each, with a mean radius of 125 nm and 0.8  $\mu$ m, respectively. The concentration of aerosols acting as CCN ( $N_{\rm CCN}$ ) is set to a constant value between the ground and 1000 m height, and it decreases exponentially up to 10,000 m, where it reaches the constant value of 0.01 cm<sup>-3</sup>. The concentration of aerosols acting as INP ( $N_{\rm INP}$ ) is homogeneous and set to a constant value. In the MID-WARM and COLD simulations, continental aerosols (ammonium sulfate, kappa = 0.61) were used as CCN, while marine aerosols (sea salt, kappa = 1.28) were used in the WARM simulations. The INP mode is composed of 61% of dust, 33% of black carbon and 6% of organic matter in all simulations (Phillips et al., 2008; Vié et al., 2016).

The choice of the non inductive charging parameterization can impact model results, both in terms of charge structure and total number of flashes (Helsdon Jr. et al., 2001; Altaratz et al., 2005; Mansell et al., 2005; Barthe and Pinty, 2007a; Fierro et al., 2006; Kuhlman et al., 2006; Tsenova et al., 2013). Both the parameterizations of Saunders and Peck (1998) and Takahashi (1978) have been widely used to simulate the electrical activity of thunderstorms. However, recent laboratory studies have shown strong similarities between the charge reversal line in Takahashi (1978) and the ones in Pereyra et al. (2000), Saunders et al. (2006) or Emersic and Saunders (2010), leading us to choose the parameterization of Takahashi (1978) for the non-inductive charge separation in this study. The inductive charging is also activated.

Concerning the lightning scheme, the fractal parameters are set to  $\chi = 2.3$  and  $L_{\chi} = 1000$  m.

## 2.3 Sensitivity tests






Since ice crystal production depends on both primary and secondary ice production processes, their contributions are tested separately. A first series of tests is carried out on CCN and INP concentrations. Simulations were performed with  $N_{\rm CCN}$  that can take five different values (500, 1000, 5000, 8000, and 10,000 cm<sup>-3</sup>), and  $N_{\rm INP}$  that can also take five different values (0.1, 1, 10, 100, and 1000 L<sup>-1</sup>). For each fixed  $N_{\rm INP}$ ,  $N_{\rm CCN}$  was varied over its five possible values resulting in a total of 25 simulations. The large range of  $N_{\rm CCN}$  and  $N_{\rm INP}$  values used in this study was inspired by the literature. Mansell and Ziegler (2013) made  $N_{\rm CCN}$  vary over 13 values between 50 and 8000 cm<sup>-3</sup>, while Tan et al. (2017) used four different values between 100 and 3000 cm<sup>-3</sup>. As for Yang et al. (2020), they used INP concentrations of 300, 800 and 1300 L<sup>-1</sup>. In the remainder of the paper, low  $N_{\rm CCN}$  refers to 500 cm<sup>-3</sup>, medium to 1000 and 5000 cm<sup>-3</sup> and high to 8000 and 10,000 cm<sup>-3</sup>. In the same way, low  $N_{\rm INP}$  corresponds to 0.1, 1 and 10 L<sup>-1</sup>, medium to 100 L<sup>-1</sup> and high to 1000 L<sup>-1</sup>. In this first set of simulations, only the HM process as a SIP mechanism is activated. For decades, two-moment schemes include a parameterization of the HM process (e.g. Ferrier, 1994; Straka and Mansell, 2005; Seifert and Beheng, 2006; Vié et al., 2016), while the CIBU and RDSF mechanisms have been only recently included in microphysics schemes (Phillips et al., 2017a, 2018; Hoarau et al., 2018; Sullivan et al., 2018; Grzegorczyk et al., 2025a) with uncertainties remaining regarding the number of fragments produced by

these processes (Grzegorczyk et al., 2025b). Moreover, CIBU and RDSF can be activated or deactivated in LIMA at the user's discretion while HM is systematically activated. Therefore, it was decided to keep HM active in these first series of simulations.

Four additional simulations are performed for each storm to analyze the impact of SIP processes on cloud electrification and lightning activity. First the CIBU process is activated in addition to the HM process. The number of fragments is randomly generated between 1 per 10 collisions and 100 per collision (Hoarau et al., 2018). Secondly, the RDSF process is activated in addition to the HM process. In the third test the HM process is disabled, resulting in a simulation in which no SIP process is considered (hereinafter referred to as NOSIP). In the last simulation, all three SIP processes are activated (hereinafter referred to as ALLSIP). In this series of simulations, aerosol concentrations representative of average aerosol conditions are used. Rose et al. (2021) have surveyed aerosol concentrations using the network of Global Atmosphere Watch (GAW) stations. Using particle number concentration in the range 100-500 nm as a proxy for potential CCN population, they showed that the potential CCN concentration ranges between a few hundreds to a few thousands particles cm<sup>-3</sup> over the continents. Mansell and Ziegler (2013) and Sun et al. (2021) used values around  $1000 \text{ cm}^{-3}$  in their modeling studies. Regarding INP concentrations, Kanji et al. (2017) showed that most studies exhibit INP concentrations between 0.5 and 50 L<sup>-1</sup> at temperatures colder than -15°C. Therefore,  $N_{\text{CCN}} = 1000 \text{ cm}^{-3}$  and  $N_{\text{INP}} = 10 \text{ L}^{-1}$  are used in all these simulations. The HM simulation where only the HM process is activated corresponds to the simulation with  $N_{\text{CCN}} = 1000 \text{ cm}^{-3}$  and  $N_{\text{INP}} = 10 \text{ L}^{-1}$  from the first set of simulations. The structure of these three storms is illustrated for this particular setup in the Supplementary Material (Figure S2).

# 3 Aerosol impact on cloud electrification and lightning activity

# 3.1 Electrical activity




Figure 2a-c represents the total flash number for each  $N_{\rm CCN}$  and  $N_{\rm INP}$  pairing during 1 hour. The lightning activity shows a large variability between the three thunderstorms, and depending on  $N_{\rm CCN}$  and  $N_{\rm INP}$ . For the same fixed values of  $N_{\rm CCN}$  and  $N_{\rm INP}$ , the three idealized cases produce a different amount of flashes during 1 h. The MID-WARM case is the most electrically active storm with a total number of flashes ranging from 625 to 4461. The WARM and COLD cases have a less intense lightning activity with the total number of flashes not exceeding 2800. In the remainder of this paper we will focus on the modification of the electrical activity and of the microphysics of each idealized case due to the sensitivity tests rather than on the differences between the three cases with the same aerosol concentration and SIP process conditions.

For all values of  $N_{\rm INP}$ , the total number of flashes is minimum for each storm when  $N_{\rm CCN}=500~{\rm cm^{-3}}$ . In general, the total number of lightning flashes tends to increase with  $N_{\rm CCN}$ , as in the WARM case at low  $N_{\rm INP}$ . However, in certain cases, threshold values of  $N_{\rm CCN}$  can be observed, beyond which the total number of lightning flashes decreases. At low  $N_{\rm INP}$ , the total number of flashes slowly decreases for  $N_{\rm CCN}>5000~{\rm cm^{-3}}$ , in the COLD case. When  $N_{\rm INP}=100~{\rm L^{-1}}$ , this threshold value for  $N_{\rm CCN}$  is observed at 5000 cm<sup>-3</sup> for the MID-WARM and COLD cases, and 8000 cm<sup>-3</sup> for the WARM case. For high  $N_{\rm INP}$ , this threshold effect is still present for the MID-WARM and the COLD cases at a lower and higher  $N_{\rm CCN}$ , respectively. However, in general, the electrical activity is less intense at high than at low or medium  $N_{\rm INP}$ .

The time of the first lightning flash for each simulation is plotted on Fig. 2d-f. The WARM and MID-WARM storms exhibit little variability compared to the COLD storm. The first flash is triggered between 18 and 27 min for the WARM storm, between 20 and 32 min for the MID-WARM storm, but between 15 and 43 min for the COLD storm. The general trend is an earlier triggering of the first flash when  $N_{\text{CCN}}$  or  $N_{\text{INP}}$  is increased.







The variability in the time of the first lightning and the total number of flashes can be mostly explained by the non-inductive charging rate in the convective zone as represented in Fig. 3. An earlier triggering of the first flash can be due to an earlier cloud electrification, a more intense non-inductive charging rate or a deeper region where the non-inductive charging occurs. At high  $N_{\rm INP}$ , cloud electrification starts 5 min earlier in each storm (not shown). In the WARM case and for  $N_{\rm INP} = 1000~{\rm L}^{-1}$ , the mean non-inductive charging rate is rather low and does not evolve too much when  $N_{\rm CCN}$  is increased (Fig. 3a) which is consistent with the lightning activity in such  $N_{\rm INP}$  and  $N_{\rm CCN}$  conditions (Fig. 2a, first line). At lower  $N_{\rm INP}$ , the non-inductive charging rate in the WARM case intensifies when  $N_{\rm CCN}$  is increased. However high charging rate occurs on a restricted altitude range at  $N_{\rm CCN} = 10,000~{\rm cm}^{-3}$  which is inline with the decrease of the total lightning flash number (Fig. 2a). It must be noted that the horizontally averaged non-inductive charging rate in the convective zone in the WARM storm does not show negative charging of the graupel. It leads to the formation of a strong positive layer of charge at low altitude with relatively low negative charges in the storm for low  $N_{\rm CCN}$  regardless of  $N_{\rm INP}$  (see Fig. S3 in Supplement). Increasing  $N_{\rm CCN}$  results in higher average positive and negative charge density located at higher altitude, and in more complex charge structures.

A similar behavior is observed for the MID-WARM case in terms of non-inductive charging rate intensity when  $N_{\rm INP}$  and  $N_{\rm CCN}$  are varied (Fig. 3, middle column). However, the negative charging of graupel is enabled for low and medium  $N_{\rm INP}$ , and for  $N_{\rm CCN}$  higher than a threshold value. This threshold value decreases from 10,000 cm<sup>-3</sup> at  $N_{\rm INP}$  = 100 L<sup>-1</sup> to 1000 cm<sup>-3</sup> for low  $N_{\rm INP}$  values. Referring to the Takahashi (1978)'s diagram, the negative charging of graupel between -10°C and -30°C occurs for cloud water content higher than 0.2-0.3 g m<sup>-3</sup> and lower than 4 g m<sup>-3</sup>. Then, the negative charging of the graupel signs the presence of significant liquid water content at cold temperatures associated with the transport of a large number of small droplets by the udpraft when  $N_{\rm CCN}$  increases. The negative charge acquired by the graupel induces a negative layer shifted toward the cloud base (see Fig. S3 in supplement) potentially changing the cloud electrical structure when  $N_{\rm CCN}$  and  $N_{\rm INP}$  vary.

Concerning the COLD case, the relatively low variability in the total number of lightning flashes (ratio of 5.2 between the minimum and maximum total number of flashes) translates into a low variability in the intensity and altitude range of the non-inductive charging rate (Fig. 3, right column). Negative charging of graupel occurs at high  $N_{\rm INP}$  or high  $N_{\rm CCN}$  values. According to the parameterization of Takahashi (1978), the negative charging of graupel occurs as soon as the temperature falls below -10°C meaning that the cloud water content exceeds  $1~{\rm g~m}^{-3}$  at this altitude.

Therefore, in general, the total flash number can be mainly explained by the amount of charge exchanged in the convective region by the non-inductive mechanism. It must be noted that the total number of flashes and the time of the first flash evolve the same way when  $N_{\rm CCN}$  is increased for low  $N_{\rm INP}$ . In the remainder of this study only the simulation with  $N_{\rm INP} = 10~{\rm L}^{-1}$  is shown as a representative for low  $N_{\rm INP}$  simulations since they have similar tendencies in their electrical and microphysicals properties.

### 3.2 Microphysical structure of the storms

In order to explain the differences in the electrical activity among the three storms under different aerosol concentration configurations, the key factors contributing to cloud electrification are analyzed. In the parameterization of Takahashi (1978) used in this study, the cloud water content (CWC) and the temperature determine the sign and amount of charge acquired by graupel particles and ice crystals. Therefore, in the following, CWC, ice crystal number concentration and graupel mass are investigated in the convective zone during the initial stage of cloud electrification. The convective region is defined as the region where the maximum vertical velocity is higher than 5 m s<sup>-1</sup> or the instantaneous precipitation rate is higher than 20 mm h<sup>-1</sup>. The initial stage of cloud electrification is defined as the first 10 min during which the absolute value of the non-inductive charging rate integrated over the volume of the convective region is greater than 0.1 C s<sup>-1</sup>.

## 3.2.1 Ice crystal concentration




Ice crystal number concentration is an essential factor for non-inductive charging: it impacts the number of collisions and the amount of charge acquired by each particle. Figure 4 shows the mean vertical profiles of ice crystal number concentration. Most profiles show two main peaks which presence and amplitude depend on aerosol concentration and storm type. These peaks are generally found around the -40°C and -5°C isotherms. To better understand the variability of ice crystal number concentration relative to storm type and aerosol concentrations, the tendencies of the ice production processes are plotted on Fig. 5 for each simulation.

Firstly, we focus on the peak concentration of ice crystals located around -40°C. At such temperature, ice crystals can be produced by heterogeneous (first row in Fig. 5) or homogeneous (second row in Fig. 5) nucleation. Logically, the production of ice crystals through heterogeneous nucleation (Fig. 5a-c) increases with  $N_{\rm INP}$  while their production through homogeneous nucleation is favored by high  $N_{\rm CCN}$  values (Fig. 5d-f). Indeed, when  $N_{\rm CCN}$  increases, a larger number of small cloud droplets are produced, transported in the updraft and part of them are available for homogeneous freezing when reaching the -35°C isotherm. Homogeneous nucleation is the most effective at low  $N_{\rm INP}$  (Fig. 5d-f). Then, increasing  $N_{\rm CCN}$  from 500 to 5000 cm<sup>-3</sup> results in a 3-4 order of magnitude increase in the ice crystal number concentration in the upper part of the cloud (Fig. 4g-i). Above  $5000 \text{ cm}^{-3}$  and regardless of  $N_{\text{INP}}$ , the mean ice crystal number concentration in the upper part of the cloud is not significantly enhanced. On the contrary, at high  $N_{\rm INP}$ , a different behavior is observed between the MID-WARM and WARM cases, and the COLD case. In the upper part of the cloud, all curves representing different  $N_{\rm CCN}$  values are almost merged for MID-WARM and WARM, while the curves for low and high  $N_{\rm CCN}$  are separated by 2 orders of magnitude next to the -40°C isotherm in the COLD case. Indeed, in the COLD case, the lack of efficiency of the warm-rain processes leading to smaller and more numerous droplets, and allows a significant quantity of supercooled water to reach the -40°C isotherm (Fig. 6c) and freeze (Fig. 5f). On the contrary, for the WARM and MID-WARM cases, in such high  $N_{\rm INP}$ , supercooled droplets are riming ice particles and are competing for water vapor with INP, leading to less droplets available for homogeneous nucleation of ice crystals (Phillips et al., 2007; van den Heever et al., 2006).

Lastly, the Hallett-Mossop process is responsible for the second peak of ice crystal number concentration close to the  $0^{\circ}$ C isotherm. For the WARM and COLD cases, the HM process rate is maximum for  $N_{\rm INP} \geq 100~{\rm L}^{-1}$  and  $N_{\rm CCN} \geq 5000~{\rm cm}^{-3}$ . For the MID-WARM case, the maximum values of the ice crystal production rate via the HM process are also obtained for  $N_{\rm INP} \geq 100~{\rm L}^{-1}$ , but for medium  $N_{\rm CCN}$  values. These differences are the result of a combination of factors that can add up or cancel each other out depending on  $N_{\rm CCN}$ ,  $N_{\rm INP}$  and the warm-phase thickness. The parameterization of the HM process in LIMA follows Beheng (1987). Accordingly, the efficiency of this process increases with the number of cloud droplets with diameter in the range 12 to 26  $\mu$ m, and the graupel mass in the region where the temperature is between -3°C and -8°C. Now, cloud droplet number concentration increases monotonically with  $N_{\rm CCN}$ , while their size decreases (not shown). At low and high  $N_{\rm CCN}$ , cloud droplets are therefore either too large or too small to be effective at rime splintering (Takahashi, 1984; Borys et al., 2003; Mansell and Ziegler, 2013). That is why the HM process is the most intense not for the highest  $N_{\rm CCN}$  but at 8000 and 1000 cm<sup>-3</sup> in the WARM and MID-WARM cases, respectively. As for the impact of  $N_{\rm INP}$  on the HM process, it is via the cloud water and graupel content, that are discussed in the next sections.

#### 3.2.2 Cloud water content



Figure 6 shows the mean vertical profiles of CWC in the convective region during the early stage of cloud electrification for the three storms and all the sensitivity studies on  $N_{\rm CCN}$  and  $N_{\rm INP}$ . CWC shows important variations with increasing  $N_{\rm CCN}$  and  $N_{\rm INP}$  in the three simulated thunderstorms. However, some general characteristics can be highlighted. In general, higher  $N_{\rm CCN}$  leads to higher CWC at each altitude, except for  $N_{\rm INP}$  = 100 L<sup>-1</sup> in the COLD case in which the highest CWC is reached for  $N_{\rm CCN}$  = 8000 cm<sup>-3</sup>. Moreover, the maximum altitude at which values of CWC higher than 0.01 g m<sup>-3</sup> can be found, tends to increase as  $N_{\rm CCN}$  rises. It is admitted that higher  $N_{\rm CCN}$  yields to higher number concentration of smaller droplets which tends to suppress collection and coalescence processes (Albrecht, 1989; Rosenfeld, 1999). These smaller cloud droplets can be transported at higher altitude where they are converted into ice crystals. The latent heat release is increased leading to stronger updrafts, increased upward transport of cloud droplets, and more CWC at higher altitudes (van den Heever et al., 2006; Sun et al., 2021).

The effect of  $N_{\rm INP}$  on the mean vertical profile of CWC in the convective region is more variable. In the WARM case and for  $N_{\rm INP} \leq 100~{\rm L}^{-1}$  (Fig. 6g and d), similar mean CWC profiles are obtained independently of  $N_{\rm CCN}$ . At the altitude of the  $0^{\circ}{\rm C}$  and the  $-15^{\circ}{\rm C}$  isotherms, CWC does not exceed 0.2 and 0.08 g m<sup>-3</sup>, respectively. In contrast, at high  $N_{\rm INP}$  (Fig. 6a), CWC increases with  $N_{\rm CCN}$ , and the altitude where it peaks is shifted upward. However, at the altitude of the  $-15^{\circ}{\rm C}$  isotherm, all curves converge to values between 0.05 and 0.1 g m<sup>-3</sup>. Finally, mean CWC higher than 0.01 g m<sup>-3</sup> can be found up to 8 to 10 km altitude. According to Takahashi (1978)'s diagram, graupel could only gain a positive charge during non-inductive charging in these conditions (Fig. 3a, 3d and 3g).

In the MID-WARM case (middle column in Fig. 6), the mean CWC profile reaches its maximum between 3 and 4 km altitude, i.e. around the 0°C isotherm, regardless of altitude,  $N_{\text{CCN}}$  and  $N_{\text{INP}}$ . For  $N_{\text{INP}} = 1000 \, \text{L}^{-1}$ , the highest mean values of CWC are observed ( $\sim 1.1 \, \text{g m}^{-3}$ ), but CWC is almost null at temperature colder than -15°C. When  $N_{\text{INP}}$  decreases, the maximum value of the mean CWC decreases, but significant values of CWC can be found at higher altitudes extending the

non-inductive charging zone above -15°C up to 7km and 9km altitude at medium and low  $N_{\rm INP}$ , respectively. According to the diagram of Takahashi (1978), due to relatively high CWC for temperatures colder than -10°C, negative charging of graupel occurs for medium  $N_{\rm INP}$  and high  $N_{\rm CCN}$  (Fig. 3e), and for low  $N_{\rm INP}$  and  $N_{\rm CCN} \ge 5000~{\rm cm}^{-3}$  (Fig. 3e).

The COLD case (Fig. 6c) shows less variability than the two other storms. For high  $N_{\rm CCN}$  values, and for any value of  $N_{\rm INP}$ , the mean CWC is between 0.1 and 0.2 g m<sup>-3</sup> at the altitude of the -30°C isotherm. The COLD case is thus favorable for negative graupel charging at relatively low altitude, between the -10°C and the -20°C isotherms as also revealed by Fig. 3c, 3f and 3i.

While the effect of varying  $N_{\text{CCN}}$  for a fixed  $N_{\text{INP}}$  is mainly the same with an increase of CWC at higher  $N_{\text{CCN}}$ , regardless the warm phase thickness, the effect of varying  $N_{\text{INP}}$  for a fixed  $N_{\text{CCN}}$  is less straightforward.

## 3.2.3 Graupel mass




The total mass of graupel in the convective zone during cloud electrification and between 0°C and -40°C is displayed in Fig. 7. The three storms show different impact of N<sub>CCN</sub> and N<sub>INP</sub> variations on graupel mass. While in the WARM case, maximum values of graupel mass are achieved for N<sub>CCN</sub> ≥ 8000 cm<sup>-3</sup> and N<sub>INP</sub> ≤ 100 L<sup>-1</sup>, in the MID-WARM case, they are obtained for N<sub>INP</sub> = 100 L<sup>-1</sup> for any value of N<sub>CCN</sub>. In the COLD case, both low N<sub>CCN</sub> and N<sub>INP</sub> values are conducive to large graupel mass. However, for each storm type, the ratio between the maximum and the minimum graupel mass is between 1.5 and 1.7. It suggests graupel mass is not a limiting ingredient for cloud electrification in these storms, but it can modulate the amplitude of the charge exchanged during the non-inductive process.

Graupel formation and growth are the result of many mixed-phase processes. Increasing  $N_{\rm CCN}$  enhances CWC in the mixed-phase region of clouds by forming more cloud droplets (Sect. 3.2.2) that can be transported above the 0°C isotherm and contribute to the riming growth of graupel. However only the WARM case shows an increase of the total mass of graupel with  $N_{\rm CCN}$  and independent of  $N_{\rm INP}$ . In the MID-WARM case, the graupel mass remains almost constant when  $N_{\rm CCN}$  varies for a fixed  $N_{\rm INP}$ . In the COLD case, higher graupel mass is linked to larger riming rates of raindrops on graupel at low  $N_{\rm CCN}$ . The cold cloud base of this storm prevents the growth of most of the raindrops to precipitation size, promoting the transport of smaller raindrops at sub-zero temperatures.

In all storms, the graupel mass decreases at high  $N_{\rm INP}$ . Indeed, graupel formation is accelerated through the rapid formation of ice crystals by heterogeneous nucleation when  $N_{\rm INP}$  increases (Fig. 5c). These crystals aggregate, then graupel mass is increased by riming, and raindrop freezing after collisions with ice crystals. At high  $N_{\rm INP}$ , graupel formation is thus accelerated, but its growth rate is limited, leading to a lower graupel mass.

### 3.3 The relationship between aerosols, microphysics and electrification

In general, increasing  $N_{\rm CCN}$  and  $N_{\rm INP}$  leads to an amplification of lightning activity due to increased ice crystal production. In this study, the enhancement of lightning activity with increasing  $N_{\rm CCN}$  varies between the three storms, with maximum enhancement factors of 11, 7, and 4 in the WARM, MID-WARM, and COLD cases, respectively. These values are of the same order of magnitude as the ones in the the literature. Sun et al. (2023) found that the total number of flashes was multiplied

by 5 when  $N_{\rm CCN}$  increased from 400 to 6,400 cm<sup>-3</sup> in a simulated multicell storm developing in a high CAPE environment. Huang et al. (2025) reported a nearly 60-fold increase of the total lightning number in a simulated squall line when the aerosol concentration increased from 400 to 4,000 cm<sup>-3</sup>. Observational studies based on AOD and lightning strikes data report similar increases in lightning activity, with enhancement factors ranging from 1.6 to 9 (Thornton et al., 2017; Naccarato et al., 2003; Proestakis et al., 2016).

This study confirms that increasing aerosol concentration leads to an amplification of lightning activity, but only up to a threshold value. Previous numerical experiments found a  $N_{\rm CCN}$  threshold around 2,000 cm<sup>-3</sup> (Mansell and Ziegler, 2013; Tan et al., 2017). We further show that the  $N_{\rm CCN}$  threshold lies in a large range of values (1,000 - 8,000 cm<sup>-3</sup>) and depends on both warm-phase thickness and  $N_{\rm INP}$ .

Increasing  $N_{\rm INP}$  naturally results in higher heterogeneous nucleation rate. However, it is less efficient to produce large concentrations of ice crystals compared to homogeneous nucleation and HM process. The HM process is shown to depend upon  $N_{\rm CCN}$  as already highlighted by Takahashi (1984), Mansell and Ziegler (2013) and Borys et al. (2003). But the relationship is less straightforward than in previous studies due to the combined effects of varying  $N_{\rm INP}$  and warm-phase thickness. A deeper warm-phase favors raindrop formation and precipitation, reducing the supercooled water content in the mixed-phase region which is essential for cloud electrification. On the contrary, a shallower warm-phase region does not provide an environment where cloud droplets can grow through collision-coalescence processes. In these conditions, smaller raindrops and cloud droplets are more easily found at sub-zero temperatures, increasing the depth of the region where the non-inductive charging can occur.

In this study, the graupel mass in the convective region during the early stage of cloud electrification was marginally impacted by aerosol concentrations compared to CWC and ice crystal concentration.

Variations in aerosol concentrations modify both the amplitude and the sign of the charge exchanged during the non-inductive process, and thus the polarity of the cloud's charge structures. Numerous studies have shown that the choice of the non-inductive process parameterization can strongly influence model results, both in terms of charge structure and number of flashes (Helsdon Jr. et al., 2001; Altaratz et al., 2005; Mansell et al., 2005; Barthe and Pinty, 2007a; Fierro et al., 2006; Kuhlman et al., 2006; Tsenova et al., 2013). Therefore, the charge structures shown in this study would be different if the Saunders and Peck (1998) parameterization was used. However, the objective of this study is not to evaluate which parameterization of the non-inductive process is the best suited for storm modeling, but rather to isolate and explore the effect of ice production on cloud electrification.

## 4 Effect of secondary ice production on cloud electrification and lightning activity

#### 4.1 Electrical activity





Figures 8 and 9 show the total number of flashes and the time of the first flash, and the total charge gained by graupel in the convective zone during cloud electrification, respectively, for all SIP-related sensitivity tests. When no SIP process is activated

(NOSIP), the total number of flashes is the lowest and the first flash is triggered the latest among all tests, for all three cases. This is due to a very low charging rate ( $\leq 1~{\rm pC\,m^{-3}\,s^{-1}}$ ) in a very small cloud depth ( $\leq 1.5~{\rm km}$ ).

Activating the HM process leads to a higher number of flashes, especially in the COLD case. In this case, the total number of flashes is doubled from 466 to 1009 between the NOSIP and the HM simulations. Additionally, the first flash is triggered 3 to 7 min earlier, and the non-inductive charging zone is clearly enhanced (Fig. 9).

Activating the CIBU process in addition to the HM process (HM+CIBU) multiplies the total number of flashes by  $\sim$ 25 for the WARM and COLD cases, and by  $\sim$ 8 for the MID-WARM case. In addition, the time of the first flash is further reduced compared to the HM simulations (between 2 and 6 min). This higher and earlier lightning activity is associated with a dramatic increase of the non-inductive charging rate up to 30 pC m<sup>-3</sup> s<sup>-1</sup> (Fig. 9).

When the RDSF process is activated in addition to the HM process (HM+RDSF), the total number of flashes increases compared to the HM simulations, for the WARM and MID-WARM cases. This enhancement is 7 times higher in the WARM case than in the MID-WARM case. In the WARM case, RDSF has a slightly higher impact than CIBU, whereas in the MID-WARM case RDSF has half the impact of CIBU. RDSF and CIBU share the same time of the first flash in these two storms (Fig. 8b). In contrast, the RDSF process does not affect the electrical activity (Fig. 8a) and the non-inductive charging rate (Fig. 9) of the COLD storm.

Finally, when all SIP processes are activated (ALLSIP), the total number of flashes is maximum and the time of the first flash is minimum for the three types of storm. The total number of flashes is multiplied by 75, 21 and 53 compared to the NOSIP simulations for the WARM, MID-WARM and COLD cases, respectively. In the WARM and MID-WARM cases, the dramatic increase in the total number of flashes is largely due to the combined and significant impact of the RDSF and CIBU processes. However, the effect of CIBU is almost 25 times, 2 times, and similar to that of RDSF in terms of total lightning activity in the COLD, MID-WARM, and WARM cases, respectively. The first flash is triggered 10 min earlier compared to the NOSIP simulations.

In general, SIP processes intensify the average density of charge and can modify the charge structure of the cloud (see Fig. S4 in Supplement). While HM and CIBU processes impact the charge structure below 10 km altitude, the effect of CIBU is more visible above 10 km altitude, in all storms.

# 410 4.2 Microphysics

## 4.2.1 Ice crystal number concentration

Figure 10 shows the SIP tendencies summed on the convective region of each storm for each simulation while Fig. 11 displays the mean vertical profiles of ice crystal number concentration for each storm and each simulation.

The HM process has the lowest tendency among the three SIP, due to the restricted range of temperature in which it is active and the relative low number of splinters produced. However, this is enough to increase the ice crystal number concentration by more than one order of magnitude between 5 and 7 km altitude in the three storms (Fig. 11). The ice crystal number concentration is also increased between 8 and 12 km altitude when the HM process is active in the WARM and MID-WARM

simulations. From Fig. 10a and 10b, this peak can be assigned to an increase of the homogeneous nucleation tendency. This is due to the conditions for cloud electrification that are met 5 min later in NOSIP, at a time when homogeneous nucleation is less active.






In the WARM case, the HM process tendency is similar for the two pairs of simulations HM and HM+CIBU ( $6.5 \times 10^9 \text{ kg}^{-1} \text{ s}^{-1}$ ), and HM+RDSF and ALLSIP ( $7.1 \text{ and } 7.2 \times 10^9 \text{ kg}^{-1} \text{ s}^{-1}$ ), meaning that RDSF has a positive impact on the HM process. The CIBU process is very efficient in producing ice crystals over the whole mixed and cold cloud depth, leading to an increase of ice crystal number concentration by around two orders of magnitude compared to the NOSIP simulation (green and blue lines in Fig. 11a). It peaks at 10 km altitude with value  $\sim 300 \text{ L}^{-1}$ . RDSF is the most efficient SIP in this storm; it induces a maximum of  $1000 \text{ L}^{-1}$  at 15 km altitude (orange line in Fig. 11a). Despite being the most active at -15°C (7.5 km in the WARM case), the RDSF process results in high  $N_i$  throughout the whole mixed and cold cloud depth, as the CIBU process, due to vertical transport. In such a deep warm cloud depth, cloud droplets can be efficiently converted into raindrops, providing a favorable environment for the RDSF process. When the three SIP processes are active (ALLSIP), they add up to produce mean ice crystal number concentration that reaches a maximum of  $1500 \text{ L}^{-1}$ .

In the MID-WARM case, the HM process increases the mean ice crystal number concentration by up to 3 orders of magnitude between 4 and 6 km altitude in the vicinity of its active temperature range, producing ice crystal number concentration up to 8  $L^{-1}$  (black line in Fig. 11b). The RDSF process increases the ice crystal number concentration by a factor 10 around 6 km altitude which is inline with its parameterization. CIBU also makes the ice crystal number concentration increase by up to a factor of 10, but over the whole mixed and cold cloud depth. As in the WARM, case, the ALLSIP simulation produces the highest mean ice crystal number concentration. Despite a stronger tendency for the RDSF process than for the CIBU process, the HM+RDSF simulation presents lower values of ice crystal concentration along the vertical profile. Indeed, the RDSF process produces a high amount of ice crystals at the early stage of the storm but becomes rapidly inactive. Grzegorczyk et al. (2025a) found a similar evolution of the RDSF process which get surpassed by the HM mechanism when the storm starts to glaciate. Actually, RDSF needs a deep warm-phase cloud depth and a moderate updraft which will help raindrops to grow and to be lifted up to the right temperature region, around -15°C (Sullivan et al., 2018). Interestingly, in the ALLSIP simulation, the RDSF process tendency is tripled compared to the HM+RDSF simulation 10. This demonstrates a positive feedback from the CIBU process; the production of additional ice crystals increases the collisions with rain drops.

Figure 10c shows that the COLD case has a particular behavior compared to the WARM and MID-WARM cases (Fig. 10a and 10b, respectively). Due to its limited warm cloud depth (less than 1.5 km thick, Fig. 1c), there is little opportunity for warm rain to form (Gupta et al., 2023), and to further participate to ice multiplication through the RDSF process (Fig. 10c). Consequently the curves of the mean ice crystal number concentration are merged in Fig. 11c for the HM and HM+RDSF simulations. In contrast, the HM and CIBU processes increase the mean ice crystal number concentration by up to a factor of 1000 in the temperature range in which they are active, i.e. between -3 and -8 $^{\circ}$ C and in the mixed-phase region, respectively. In the HM+CIBU and ALLSIP simulations, the mean ice crystal number concentration reaches 500 L $^{-1}$  at 11 km altitude.

#### 4.2.2 Cloud water content





Figure 12 shows the mean vertical profiles of CWC during cloud electrification. In the WARM case, the NOSIP simulation produces the lowest CWC. It reaches a maximum of  $0.08~{\rm g\,m^{-3}}$  near the  $0^{\circ}{\rm C}$  isotherm against a maximum of  $0.15~{\rm g\,m^{-3}}$  in all simulations where SIP processes are activated. As soon as one SIP process is activated, all mean vertical profiles of CWC are merged in the WARM and MID-WARM cases. In the MID-WARM case, in the altitude range between the  $10^{\circ}{\rm C}$  and  $-10^{\circ}{\rm C}$  isotherms, CWC is higher in the NOSIP simulation than in all simulations where SIP processes are activated. At temperatures colder than  $-10^{\circ}{\rm C}$ , CWC exponentially decreases in the NOSIP simulation, while higher CWC are found at higher altitude when SIP processes are considered.

It is important to note that the beginning of the electrification period may be different in the different sensitivity studies. As SIP processes accelerate the formation of ice particles, cloud electrification starts 5 min earlier as soon as one SIP process is activated compared to the NOSIP simulation, in the WARM and MID-WARM storms. When the mean vertical profile of CWC is computed at the same time period as the one when SIP processes are activated (not shown), CWC is lower when SIP processes are considered. This is in agreement with previous numerical studies of SIP impact (Zhao and Liu, 2022; Grzegorczyk et al., 2025a). Indeed, SIP processes are sink of CWC through the riming of snow/aggregates and graupel. The COLD case does not show any impact of the SIP processes on the average CWC profile in the early cloud electrification stage. The SIP processes do not change the timing of cloud electrification onset in the COLD storm. As cloud electrification starts during the development stage of the cloud, SIP processes have not yet consumed CWC. Despite the presence of significant CWC in the mixed phase region above the 0°C isotherm in NOSIP simulations across all storms, the non-inductive charging process only occurs at high altitude (between 7.5 and 11 km), where ice crystals are available (Fig. 9 and Fig. 11).

#### 470 **4.2.3** Graupel mass

As in Sect. 3.2.3, only slight changes are found in the graupel mass in the SIP series of simulations. In general, the total mass of graupel is higher when no SIP process is activated. The total graupel mass decreases from  $16 \times 10^8$  to  $14 \times 10^8$  kg, and from  $12 \times 10^8$  to  $9.8 \times 10^8$  kg in the WARM and MID-WARM cases, respectively, as soon as the HM process is taken into account. In the COLD case, the total graupel mass only varies between  $6.9 \times 10^8$  and  $6.5 \times 10^8$  kg, with the maximum value for the NOSIP simulation. However, in this case study, only the inclusion of the CIBU process reduces the total mass of graupel.

SIP processes reduce graupel mass through two different pathways. The HM and RDSF processes directly consume graupel to form ice crystals while the CIBU parameterization of Hoarau et al. (2018) considers that splinters originate from breaking aggregates during collisions with graupel. The reduction of graupel mass when SIP processes are activated can also be attributed to the competition for CWC, which is shared between the riming of snow particles and the riming of graupel. SIP mechanisms produce numerous ice crystals, which can aggregate or grow into snow particles by water vapor deposition. As a result, cloud droplets increasingly rime onto snow/aggregates at the expense of graupel riming growth.

### 4.2.4 The relationship between SIP processes, microphysics and electrification







In the NOSIP simulations, only homogeneous and heterogeneous nucleation produce ice crystals resulting in low ice crystal number concentration at warm temperatures which limits the non-inductive charging rates. It can explain the late triggering of flashes and the low lightning activity in all storms.

Activating SIP processes enhances the ice crystal number concentration and the lightning activity, with an impact 5 times greater than that of aerosol concentration in terms of the number of flashes. This is lower than the 100-fold flash rate increase deduced from Huang et al. (2025) when multiple SIP processes are activated, especially under high CCN concentrations. Using the WRF model, Yang et al. (2024) also found that when SIP processes are taken into account in a simulation of a cold-season thundesrtorm, collisions between graupel and ice crystals are enhanced leading to an increase of the vertical electric field and flash rate.

Numerical studies consistently highlight the dominant role of SIP processes over primary ice production especially in the mixed phase region (Huang et al., 2022; Grzegorczyk et al., 2025a). However, SIP efficiency can vary with microphysical conditions. For instance, Zhao and Liu (2022) found reduced SIP rates when using a stronger primary ice nucleation parameterization: cloud glaciation is accelerated, and rain and graupel formation is reduced which inhibits SIP processes. In the present study, SIP sensitivity was tested using only one set of  $N_{\rm CCN}$  and  $N_{\rm INP}$ , and the sensitivity to SIP parameterization has not been explored. Prior work (e.g. Mansell and Ziegler, 2013) has shown that different HM parameterizations significantly influence electrification.

In general, all SIP processes produce a high ice crystal number concentration at altitudes lower than those at which homogeneous and heterogeneous nucleation occurs. This enables the cohabitation between CWC, high ice crystal number concentration and graupel particles which results in more intense cloud electrification. However, each SIP mechanism has a different impact according to the cloud base temperature of the storm. HM and CIBU processes enhance cloud electrification and lightning activity in every storm. Activating the HM process makes the first flash to be triggered approximately 5 min earlier. When combined with CIBU or RDSF, it can advance the triggering of the first flash by up to 10 min. The RDSF process requires specific and challenging conditions to take place. In the COLD case, this process is not active due to the lack of raindrop formation. In the MID-WARM storm, the RDSF process can take place and produce a large number of ice crystals, but it becomes rapidly inactive. In contrast, in the WARM storm, the RDSF process is very efficient in producing secondary ice crystals and is active during the whole cloud electrification period thanks to a constant raindrops supply and efficient updraft.

SIP processes also impact the mean CWC vertical profile and the graupel mass, though to a lesser extent within the short window of cloud electrification defined in this study. The main differences observed on these two parameters are due to different cloud electrification onsets and to the enhanced production of snow/aggregate particles that can grow by riming of cloud droplets at the expense of graupel.

The weak sensitivity of CWC to SIP processes may result from several factors. Data sampling during the electrification period limits the detection of differences, which occur more significantly during the storm's mature stage. Another possible explanation is the use of a saturation adjustment scheme in LIMA. This adjustment, applied after all other microphysical

processes, forces the environment to reach a strict equilibrium at water saturation at the end of each time step. The use of saturation adjustment can overestimate condensate mass and enhance rain formation, reducing supercooled water (Khain et al., 2015; Zhang et al., 2021). In contrast, studies showing stronger CWC responses to SIP does not use saturation adjustment schemes (Phillips and Patade, 2022; Grzegorczyk et al., 2025b; Huang et al., 2025). Additionally, in the version of LIMA used in this study, snow and graupel number concentrations are not prognostic, potentially accelerating their formation and depleting liquid and small ice species in comparison with a full two-moment version of LIMA (Taufour et al., 2024).

## 5 Conclusions







Three idealized thunderstorms that differed by their warm-phase cloud thickness were simulated in order to assess the influence of ice production processes on cloud electrification and lightning activity. This was done using the cloud-resolving model Meso-NH with the quasi two-moment microphysics scheme LIMA coupled with the explicit electrical scheme CELLS. A first set of simulations was performed by simultaneously varying the number concentration of aerosols acting as INP and CCN. A second set of simulations was conducted in which three different SIP processes were alternately active or deactivated. Our results indicate that both aerosol concentration and SIP processes alter the cloud microphysics and the subsequent electrical activity. Several effects can be observed: a delay in the onset of cloud electrification and in the triggering of the first lightning flash, as well as a change in the total number of flashes.

Sensitivity tests on aerosol concentration show that an increase in  $N_{\rm CCN}$  and  $N_{\rm INP}$  generally enhances lightning activity. Aerosol concentrations affect cloud electrification by modulating the vertical profiles of CWC and ice crystal number concentration. A higher  $N_{\rm CCN}$  leads to a greater CWC, expanding the mixed-phase region of the cloud, while a higher  $N_{\rm INP}$  depletes CWC at high altitude, altering both the sign and magnitude of the charge exchanged during the non-inductive charging mechanism.  $N_{\rm INP}$  favours an accelerated production of rimed particles as a result of a sequence of microphysical processes. Aerosols also control homogeneous nucleation, which is dominant at high  $N_{\rm CCN}$  and low  $N_{\rm INP}$ . Therefore, by increasing the ice crystal number concentration, aerosol concentration controls the number of ice crystal-graupel collisions, thereby influencing the amount of charge exchanged at each grid point. Despite a similar onset of cloud electrification, the triggering time of the first flash can differ according to the intensity of charge separation by the non-inductive process.

However, the increase of lightning activity with aerosol concentration can be monotonic or up to a specific threshold ( $N_{\rm INP}$ :  $100\,{\rm L}^{-1}$ ;  $N_{\rm CCN}$ :  $1000-8000\,{\rm cm}^{-3}$ ). The total number of flashes decreases beyond these thresholds. Previous observational and numerical modeling studies have also found an enhancement of lightning activity with high aerosol loading up to a threshold value. Numerical modeling studies identified a CCN threshold around  $2000\,{\rm cm}^{-3}$  (Altaratz et al., 2010; Shi et al., 2020; Mansell and Ziegler, 2013; Tan et al., 2017) which is of the same order of magnitude as our study. Mansell and Ziegler (2013) attributed the decrease of lightning activity with  $N_{\rm CCN}$  to the HM process, while Tan et al. (2017) hypothesized that vapor competition leads to a decrease in ice crystal size and mixing ratio. Most numerical modeling studies have focused only on  $N_{\rm CCN}$  or  $N_{\rm INP}$ . Our findings highlight a complex interaction between CCN and INP, and both particles have to be taken into account to understand the aerosol impact on cloud electrification and lightning activity.

Sensitivity tests on SIP processes (rime splintering, raindrop shattering by freezing, and collision ice breakup) demonstrated that they are essential to produce high ice crystal number concentration, especially at low altitudes where primary ice production does not occur. When no SIP process is activated and regardless of the simulated storm, the ice crystal number concentration remains low, resulting in a weak cloud electrification and lightning activity. When they are active, the intensity of each process depends on the thickness of the cloud's warm-phase. A thick cloud's warm-phase region favors the growth of cloud droplets and their conversion to raindrops, enabling raindrop shattering freezing afterwards. On the contrary, a thinner cloud's warm-phase region creates an environment with fewer raindrops and more supercooled cloud droplets aloft conducive to the HM process. The CIBU process is active regardless of the cloud base temperature. This differentiated effect of SIP processes on cloud electrification is consistent with Phillips and Patade (2022) results for a cold-base thunderstorm in which HM and RDSF are almost inactive.

Comparing the impact of aerosol and SIP processes on cloud microphysics and electrification, it is clear that both ice production pathways are essential for cloud electrification through the non-inductive charging mechanism. However, SIP processes have a more important impact on the cloud electrification and the resulting lightning activity. While variations in the aerosol concentration can increase the total number of flashes by up to an order of magnitude, activating the SIP processes can multiply the total number of flashes by 50 in some cases. It is also important to note that the relationship between aerosol concentration, SIP processes, and cloud electrification is complex and varies depending on the cloud base temperature. This study highlights the importance of taking into account the formation of ice crystals via SIPs, as this largely determines the conditions required for the non-inductive mechanism to take effect. However, uncertainties arise from the parameterizations of the SIP processes in the model. For example, several studies have proposed a more complex parameterization of the CIBU process by including dependence on physical parameters (Phillips et al., 2017b; Grzegorczyk et al., 2025a). Additionally, there is still no consensus on the parameterization of the non-inductive process, and several existing parameterizations should be tested.

The next step will be to simulate a thunderstorm observed during the EXAEDRE (EXploiting new Atmospheric Electricity Data for Research and the Environment) field campaign that took place in Corsica in 2018. This campaign offers many observations of cloud microphysics and electric activity, including data from operational weather radar, from a suite of airborne microphysics probes and the airborne 95 GHz Doppler cloud radar RASTA onboard the French Falcon research aircraft, and from the SAETTA network (Coquillat et al., 2019). It provides a robust database for comparison with the numerical simulations. This study will focus on SIP processes and will be crucial for improving our understanding of SIP processes and their role in cloud electrification, as well as validating the findings presented in this study.

Code availability. Version 5-7 of Meso-NH is under the CeCILL-C license agreement and freely available at http://mesonh.aero.obs-mip.fr/mesonh57.

Author contributions. IV: Writing – original draft, Methodology, Investigation, Formal analysis, Visualization. CB: Writing - Review & Editing, Conceptualization, Software, Methodology, Investigation, Validation, Funding acquisition, Supervision

Competing interests. The contact author has declared that none of the authors has any competing interests.

Acknowledgements. This work supported by Agence Nationale de la Recherche (ANR) of France under grant ANR-21-CE01-0006 (ANR ICCARE). Computations were performed on the local computer cluster of Laboratoire d'Aérologie.

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

Figure 1. Thickness of the warm, mixed and cold-phase regions of the WARM (a), MID-WARM (b) and COLD (c) storms during the cloud electrification period.

Figure 2. Normalized total number of flashes in colors (first line, a-c) and normalized time of the first flash (colors) (second line, d-f) as a function of  $N_{\text{CCN}}$  and  $N_{\text{INP}}$ , for the WARM (left column), MID-WARM (middle column) and COLD (right column) simulations. The normalization is obtained by substracting the minimum and dividing by the difference between the maximum and minimum. The text in each grid box corresponds to the total number of lightning flashes (first line) and the time of the first flash in minutes (second line) in the corresponding simulation.

Figure 3. Non-inductive charge separation rate on graupel and ice crystals averaged in the convective zone during cloud electrification as a function of  $N_{\text{CCN}}$ , for  $N_{\text{INP}} = 1000 \text{ L}^{-1}$  (top line),  $N_{\text{INP}} = 100 \text{ L}^{-1}$  (middle line) and low  $N_{\text{INP}}$  (bottom line) of the WARM (left column), MID-WARM (center column), and COLD (right column) simulations. A positive (negative) value corresponds to a positive (negative) charge gained by the graupel particle after collision with an ice crystal.

Figure 4. Mean vertical profiles of ice crystal number concentration ( $\rm L^{-1}$ ) for  $N_{\rm INP}$  = 1000  $\rm L^{-1}$  (top line),  $N_{\rm INP}$  = 100  $\rm L^{-1}$  (middle line) and low  $N_{\rm INP}$  (bottom line) in the convective region during cloud electrification of the WARM (left column), MID-WARM (center column), and COLD (right column) cases. In each panel, the blue, black, green, orange and pink curves correspond to the mean vertical profiles of ice crystal number concentration for  $N_{\rm CCN}$  = 500, 1000, 5000, 8000 and 10,000 cm<sup>-3</sup>, respectively. The 0°C and -40°C isotherms are plotted with black dashed lines.

**Figure 5.** Normalized tendencies of the three ice production processes summed on the vertical (colors) of the WARM (left column), MID-WARM (center column) and COLD (right column) simulations: heterogeneous nucleation (first line), homogeneous nucleation (middle line) and Hallett-Mossop process (bottom line). The text in each grid box corresponds to the ice production processes tendencies summed on the vertical  $(\times 10^9 \text{ s}^{-1})$  in the corresponding simulation.

**Figure 6.** Mean vertical profiles of cloud water content (CWC, in gm $^{-3}$ ) in the convective region during cloud electrification as a function of  $N_{\rm CCN}$ , for  $N_{\rm INP} = 1000~{\rm L}^{-1}$  (top line),  $N_{\rm INP} = 100~{\rm L}^{-1}$  (middle line) and low  $N_{\rm INP}$  (bottom line) of the WARM (left column), MID-WARM (center column), and COLD (right column) simulations. The  $0^{\circ}{\rm C}$ ,  $-10^{\circ}{\rm C}$ ,  $-20^{\circ}{\rm C}$ ,  $-30^{\circ}{\rm C}$ , and  $-40^{\circ}{\rm C}$  isotherms are plotted with black dashed lines.

Figure 7. Normalized total mass of graupel (colors) in the convective zone during cloud electrification between  $0^{\circ}$ C and  $-40^{\circ}$ C in the WARM (a), MID-WARM (b) and COLD (c) simulations as a function of  $N_{\rm INP}$  and  $N_{\rm CCN}$ . The text in each grid box corresponds to the total mass of graupel ( $\times 10^{8}$  kg) in the corresponding simulation.

**Figure 8.** Normalized total number of flashes (a) and normalized time of the first flash (b), in colors, as a function of the SIP processes activated (NOSIP, HM, HM+CIBU, HM+RDSF, ALLSIP) and of the storm type (WARM, MID-WARM, COLD). The number in each grid box corresponds to (a) the total number of lightning flashes and (b) to the time of the first flash (min).

**Figure 9.** Non-inductive charge separation rate between graupel and ice crystals summed in the convective zone during cloud electrification as a function of the SIP processes activated (NOSIP, HM, HM+CIBU, HM+RDSF, ALLSIP) and of the storm type (WARM, MID-WARM, COLD). A positive (negative) value corresponds to a positive (negative) charge gained by the graupel particle after collision with an ice crystal.

Figure 10. Normalized ice production rate of the SIP processes summed on the vertical (colors) of the (a) WARM, (b) MID-WARM and (c) COLD cases for all sensitivity tests about SIP processes. The text in each grid box corresponds to the ice production processes tendencies summed on the vertical ( $\times 10^9 \text{ s}^{-1}$ ) in the corresponding simulation.

Figure 11. Mean vertical profiles of ice crystal number concentration ( $L^{-1}$ ) in the convective region during cloud electrification of the (a) WARM, (b) MID-WARM, and (c) COLD cases. In each panel, the blue, black, green, orange and pink curves correspond to the mean vertical profiles of ice crystal number concentration for the NOSIP, HM, HM+RDSF, HM+CIBU and ALLSIP tests, respectively. The 0 and -40 $^{\circ}$ C isotherms are plotted with black dashed lines.

**Figure 12.** Mean vertical profiles of CWC (g.m $^{-3}$ ) in the convective region during cloud electrification of the WARM (la), MID-WARM (b), and COLD (c) simulations. In each panel, the blue, black, green, orange and pink curves correspond to the mean vertical profiles of CWC for each SIP sensitivity tests. The  $0^{\circ}$ C,  $-10^{\circ}$ C,  $-20^{\circ}$ C,  $-30^{\circ}$ C, and  $-40^{\circ}$ C isotherms are plotted with black dashed lines.

Figure 13. Total mass of graupel in the convective zone between -40°C and 0°C in the WARM (a), MID-WARM (b) and COLD (c) simulation. The text in each grid box corresponds to the total mass of graupel ( $\times 10^8$  kg) in the corresponding simulation.