# Peer review of "Distinct effects of several ice production processes on thunderstorm electrification and lightning activity"

_EGUsphere, 2025_

## Referee Comment (RC2)

Summary:

The manuscript "Distinct effects of several ice production processes on thunderstorm electrification and lightning activity" simulates three idealized storms, specified by the cloud base temperature and depth of the warm-phase layer, to assess the influence of aerosol concentration (CCN and INP) and three SIP processes (Hallett-Mossop rime splintering, raindrop shattering by freezing, and collision ice breakup) on cloud microphysics (ice crystal number concentration, cloud water content, graupel mass) and on electrification/lightning activity (charging rate on graupel, total number of flashes, and time of the first flash). The main results include an increase in lighting activity with the increase of CCN concentration up to a threshold as found in previous studies, but here it is shown that this threshold value varied depending on the INP concentration and type of storm. Each SIP process impacted the cloud electrification and lighting activity differently depending on the thickness of the cloud's warm-phase. The results also highlight that activating SIP processes in the simulations impacted more dramatically the lightning activity than varying/adjusting aerosol concentrations (CCN or INP). In general, the study is well-structured and presents valuable and relevant contributions within the scope of ACP, but there are some inconsistencies mainly in sections 3 and 4. My comments are included below.

General Comments:

- **Aerosol concentration and SIP process for control run**
  In line 176, it is mentioned that the aerosol concentrations were kept constant at $N\_CCN = 1000$ cm^-3 and $N\_INP = 10$ L^-1 when analyzing the impact of the SIP processes. Why were these values chosen? Was there an additional evaluation to arrive at these values? Was this choice made based on a paper? If so, I recommend including the citation. CCN concentration of 1000 cm^-3 could be considered part of the high range of CCN concentration (Mansell and Ziegler, 2013). If not, I suggest including, mentioning or highlighting what would be realistic values or range of values for the three types of storms, since the chosen sensitivity range spans a more extensive range not explored by other studies as mentioned in lines 165-174. Additionally, in line 174, why only the HM process is activated in the first set of simulation varying aerosol concentrations? Could the reasoning for this decision also be included?

- **Charge separation parameterization**
  In line 163, the authors state that "... the non-inductive charge separation is parameterized following Takahashi (1978)..." Although it is mentioned in line 528 "... there is still no consensus on the parameterization of the non-inductive

process, and several existing parameterizations should be tested." I expected the manuscript to provide more detail on the implementation of this parameterization and to discuss the potential implications its selection may have on the results. This is particularly important given that the Saunders and Peck (1998) scheme is widely used and has been shown to also successfully reproduce inverted-polarity charge structures, as demonstrated by for example by Kuhlman et al. (2006).

- **Charge density instead of just the charging rate on graupel**
  Figures 3 and 9 only present the charging rate on graupel, but this information alone does not provide a clear indication of the storm's overall charge structure. I would strongly suggest providing cross-sectional plots of the charge density to reduce ambiguity in the interpretation and validation of the results. Additionally, Figure 1 presents only the thickness of the warm, mixed and cold-phase regions of the three idealized storms. It would be beneficial to include additional context of the simulation results such as plots of the simulated radar reflectivity to illustrate how these storms evolve and to better connect to the idealized setups.

- **Comparison with Phillips and Patade (2022)**
  The results for the cold case are compared with those of Phillips and Patade (2022), showing consistency on the importance of the CIBU process, as noted in line 516: "This is consistent with Phillips and Patade (2022) results for a cold-base thunderstorm in which HM and RDSF are almost inactive."
  There are more details in the introduction from Phillips and Patade (2022) and the effect of CIBU on CWC in line 66 "Phillips and Patade (2022) found that the most active SIP process was breakup during ice-ice collisions. This process, acting as a sink of liquid water content, has the ability to alter the polarity of the charge Graupel acquires and, consequently, the electric charge structure."
  However, in line 437 and referring to figure 12 the manuscript states that: "The COLD case does not show any impact of the SIP processes on the average CWC profile in the early cloud electrification stage... As cloud electrification starts during the development stage of the cloud, SIP processes have not yet consumed CWC." The comparison as currently presented appears to lack consistency. I recommend revising the text and revisiting the simulation/analysis to address potential contradictions of the results also within the manuscript and ensure a clearer discussion.

Specific Comments:

**Abstract**
- Line 13: What impact on electrification is this referring to? Is it regarding the polarity, the charge magnitude, number of flashes, …?

**1 Introduction**
- Suggest include citations for the sentences starting in lines 20 and 21.

**2.1.1 Microphysical scheme**
- In lines 101 and 105, the authors introduce abbreviations for the SIP processes: collisional ice break-up as CIBU and raindrop shattering freezing as RDSF. But in line 99, there is no mention of the abbreviation of the Hallett-Mossop process as HM. Additionally to maintain consistency, in line 315 and 505, this process is referred to as rime splintering, when throughout the manuscript HM process has been used. This term could be introduced in line 99 as well.

- In lines 108-118, the manuscript provides implemented equations, expressions and values for the RDSF process. But the same treatment is not given to the other SIP processes HM and CIBU. Is there a reason for expanding the explanation just for RDSF and not the other processes? Was the RDSF implementation different from the cited studies?

- The units for INP concentrations are given in $L^{-1}$, but in line 172, a reference from concentrations used in another study are given in $cm^{-3}$. Writing the concentrations in the same units would help the reader to compare the range and values considered.

**Results: Sections 3 and 4**
- Recommend maintaining a structure in the results sections 3 and 4.
  In section 3, it is presented the following subsections:
  3 Aerosol impact on cloud electrification and lightning activity
  3.1 Electrical activity
  3.2 Microphysical structure of the storms
  3.2.1 Cloud water content
  3.2.2 Ice crystal concentration
  3.2.3 Graupel mass
  3.3 The relationship between aerosols, microphysics and electrification

  In section 4, they are:

4 Effect of secondary ice production on cloud electrification and lightning activity
4.1 Electrical activity
4.2 Microphysics
4.2.1 Ice crystal number concentration
4.2.2 Cloud water content
4.2.3 Graupel mass
4.2.4 The relationship between SIP processes, microphysics and electrification

So, the subsection titles and the order they appeared are modified from what was in section 3. Recommend keeping this consistent.

● In line 190: "... we will focus on the modification of the electrical activity and of the microphysics of each idealized case due to the sensitivity tests rather than on the differences between the three cases with the same aerosol concentration and SIP process conditions." But, in line 373 the results are compared across storms under the same set of conditions: "This enhancement is 7 times higher in the WARM case than in the MID-WARM case. " How much are their respective increases compared to just HM or HM+CIBU?

● In line 268, when referring to the Takahashi diagram, I would suggest citing the paper, since there are a couple of Takahashi's papers in the References section.

● There are several mentions of high and low values for $N_{CCN}$ and $N_{INP}$ but the range is only specified later in the section. I would suggest making it more clear at the beginning of the section or on the sensitivity test section the ranges for low, medium and high $N_{CCN}$ and $N_{INP}$.

● For the warm case, what is the range that the HM process is maximum/most intense, since the following sentences seem to disagree? In line 315: "That is why the HM process is the most intense for intermediate values of $N_{CCN}$ in the WARM and MID-WARM cases." But in line 309: "For the WARM and COLD cases, the HM process rate is maximum for high $N_{INP}$ ($\geq 100$ L$^{-1}$) and high $N_{CCN}$ ($\geq 5000$ cm$^{-3}$)."

● Line 325: "It suggests graupel mass is not a limiting ingredient for cloud electrification, but it can modulate the amplitude of the charge exchanged during the non-inductive process." I would recommend explaining this better as it is not clear to me the results are suggesting this.

- There are numerous instances where the word "whatever" is used. I would recommend replacing it with "regardless of" or "independent of".

- Line 338: "The formation is accelerated but the intensity is weaker leading to a lower graupel mass at high N_INP." The intensity of what is being referenced here?

- Lines 378-380. These sentences could be combined to avoid repetition.

- Line 395: "In the WARM case, the HM process tendency is identical for the two pairs of simulations HM and HM+CIBU (6.5 x 10^9 kg^−1 s^−1), and HM+RDSF and ALLSIP (7.1 and 7.2 x 10^9 kg^−1 s^−1)..." 7.1 and 7.2 are not identical values.

- Is the result in line 397: "The CIBU process is very efficient in producing ice crystals over the whole mixed and cold cloud depth, leading to an increase of ice crystal number concentration by around two orders of magnitude (green and blue lines in Fig. 11a)." in comparison to NOSIP or HM simulation?

- Line 399: "RDSF is the most efficient SIP in this storm; it induces a maximum of 1000 L^−1 (orange line in Fig. 11a)." What altitude and/or temperature does this correspond to?

- Line 400: "Despite being the most active at -15degC, the RDSF process results in high N_i throughout the whole mixed and cold cloud depth..." There is not an isotherm line for -15 degC, so what altitude does it correspond to?

- Line 402: "When the three SIP processes are active (ALLSIP), they add up to produce mean ice crystal number concentration that reaches a maximum of 1500 L^−1." Is this maximum ice crystal concentration at the same altitude of the 1000 L^_1 peak for process RDSF (line 399)?

- Is "the HM+RDSF simulation presents lower values of ice crystal concentration along the vertical profile" in line 410 a comparison to the lower values in the HM+CIBU simulations?

- Line 413: "Actually, RDSF needs a deep warm-phase cloud depth and a moderate updraft which will help raindrops to grow and to be lifted up to the right temperature region (Sullivan et al., 2018)" what is the right temperature region?

- Line 414: "Interestingly, in the ALLSIP simulation, the RDSF process 415 tendency is tripled compared to the HM+RDSF simulation." This refers to figure 10b, right? Add it here.

- Line 421: They increase the mean ice crystal number concentration by up to a factor of 1000 in the temperature range in which they are active." What is this temperature range?

- Line 428: "In the MID-WARM case, CWC is higher in the NOSIP simulation than in all simulations where SIP processes are activated near the 0degC isotherm." It looks like it is activated until close to -10degC isotherm.

- Line 439: "... the non-inductive charging process only occurs at high altitude (between 7.5 and 11 km), where ice crystals are available…" Figure 9 shows charge separation occurring for ALLSIP simulation from 5 km altitude.

- In line 473, what does "different cloud electrification onsets" mean?

- Do the plots in Figure 4 for $N\_CCN = 1000$ cm^-3 (black line), $N\_INP = 10$ L^_1 (third row) match the ones for Figure 12 with HM process (black line)? Could this correspondence of control cases be included in the manuscript?

**5 Conclusions**
- Line 491: "As for the HM process, it is maximum at intermediate or high $N\_CCN$ levels (1000 - 10,000 cm^−3)" Is maximum at producing ice crystals? Also, inconsistent definition for the high range of $N\_CCN$ in line 308.

- Lines 497-500. Combine sentences to avoid repetition.

- Line 500: "Mansell and Ziegler (2013) attributed the decrease of lightning activity with NCCN to the HM process…" A decrease is observed once the threshold value is exceeded?

- Line 503: "Thus, both particles has to be taken into account to ..." Adjust to: Both aerosol particles have to be …

---

## Author Comment (AC1)

**Reply on RC1**
* * *
**RC1:** This study presents results from a suite of thunderstorm simulations in which CCN concentrations, INP concentrations have been varied and in which SIP mechanisms have been activated or deactivated. The authors focus on the response of the lightning activity to these choices and attempt to understand the response through the analysis of the storms' microphysical properties and process rates. Three thunderstorm cases are analyzed. It is a very detailed study. The results regarding the response to CCN and INP concentrations appear to be consistent with previous studies although the present study is perhaps somewhat more robust in that it examines multiple cases. The testing of SIP mechanisms appears to be a more novel aspect of the study and here their results are not entirely consistent with the few other studies that exist. Overall I think the study has potential to be a useful contribution to the community but I do have some major questions about the results.

We thank the reviewer for his/her time and efforts in reviewing our manuscript. The responses to his/her comments are addressed below.

1. The second part of the study regarding SIP mechanisms seems to make the first part of the study (and previous studies regarding CCN/INP concentrations) potentially irrelevant. The authors show that the inclusion of additional SIP mechanisms increasing the lightning flash count by 15-50x – a substantially larger increase than was obtained by varying CCN and INP concentrations. Assuming that the inclusion of SIP mechanisms leads to a more realistic simulation, then how meaningful are the results of the CCN/INP tests with only HM? I would guess that with all SIP mechanisms included, the sensitivity to INP would vanish and perhaps the sensitivity to CCN would also be diminished?

You are right. Our study clearly shows that the SIP effect dominates ice crystal production and cloud electrification over the aerosol effect. But this result must be qualified in view of the assumptions made in the simulations.
In the second part of our study dealing with the effect of SIP on cloud electrification, only one set of CCN and INP number concentration was used. However, Hoarau et al. (2018) have shown that varying the initial number concentration of IFN may modify the cloud ice concentration by up to one order of magnitude. Huang et al. (2025) found a maximum 100-fold increase in flash rate with four different SIP activated (the ice sublimation breakup process is used in addition to the three processes studied in our manuscript). This factor is estimated from their Figure 17. Huang et al. (2025) have also shown different behavior in the total flash rate at low (400 cm$^{-3}$) and high (4000 cm$^{-3}$) CCN concentration with and without the four SIP activated. They observed a higher enhancement of the flash rate with SIP processes at higher CCN concentration.
Moreover, only one parameterization of each SIP process was used in our study. In Mansell and Ziegler (2013), a test is carried out on the formulation of the Hallet-Mossop (HM) process. Two different parameterizations are tested: one set of simulations used the parameterization of Ziegler et al. (1986), while the second set of simulations used the one of Hallett and Mossop (1974). Both parameterizations operate in the same range of temperature, between -3 and -8°C. On Figure 9 of Mansell and Ziegler (2013), we can clearly see the large discrepancy in terms of flash density depending on the formulation of the HM process. A factor 8 in terms of total lightning is estimated between the two

simulations when the number concentration of CCN is fixed at 200 cm$^{-3}$. The difference in terms of total lightning reaches a factor 20 when the CCN number concentration is 5000 cm$^{-3}$.

There is no doubt that SIPs play an important role in cloud electrification, but large uncertainties remain in their parameterizations used in microphysics schemes (Field et al., 2017; Han et al., 2024, Grzegorczyk et al., 2025). As stressed in the conclusion, the simulation of a real case with available in situ measurements will be useful to constrain the model and to better understand the physical processes at play in cloud electrification.

A paragraph has been added in Section 4.2.4 to discuss these uncertainties:

*"Activating SIP processes enhances the ice crystal number concentration and the lightning activity, with an impact 5 times greater than that of aerosol concentration in terms of the number of flashes. This is lower than the 100-fold flash rate increase deduced from Huang et al. (2025) when multiple SIP processes are activated, especially under high CCN concentrations. Numerical studies consistently highlight the dominant role of SIP processes over primary ice production especially in the mixed phase region (Huang et al., 2022; Grzegorczyk et al., 2025). However, SIP efficiency can vary with microphysical conditions. For instance, Zhao and Liu (2022) found reduced SIP rates when using a stronger primary ice nucleation parameterization: cloud glaciation is accelerated, and rain and graupel formation is reduced which inhibits SIP processes. In the present study, SIP sensitivity was tested using only one set of $N_{CCN}$ and $N_{INP}$, and the sensitivity to SIP parameterization has not been explored. Prior work (e.g. Mansell and Ziegler, 2013) has shown that different HM parameterizations significantly influence electrification."*

2. I am very surprised by the near total lack of sensitivity of the CWC profiles to SIP mechanisms (aside from NOSIP). It's not just that the CWC profiles are similar, they are virtually identical. Assuming that this is not an outright error, could it potentially be due to the parameterization of ice crystal collisions or properties in LIMA? For example, perhaps LIMA has a minimum ice crystal size that is being met and so all simulations have the same crystal size despite differences in concentration. Or there is some hard-coded limiter in the collision rate with cloud droplets? Or is there no longer a mixed-phase region? It is just very hard to explain why orders of magnitude differences in the ice crystal number concentration should have absolutely no impact on the cloud water content. It also seems potentially inconsistent with previous discussion of how INP concentrations impact CWC (for a fixed CCN concentration). Why should HM-only with INP variation impact CWC while HM+other SIP should not?

Several factors might be responsible for the lack of sensitivity of CWC with the SIP processes activated. Firstly, it partly results from the data being sampled during the electrification period defined in this study. At the mature stage, the simulations start to diverge; the ALLSIP simulation having the least CWC. However the differences remain small compared to the stronger sensitivity of CWC to aerosol concentrations.

A possible explanation for this weak response is the use of a saturation adjustment scheme in LIMA. This adjustment, applied after all other microphysical processes, forces the environment to reach a strict equilibrium at water saturation at the end of each time step, by condensing water vapor or evaporating cloud droplets depending on whether the air is supersaturated or sub-saturated. In a general way, Khain et al. (2015) examined the bin and

bulk parameterizations of microphysics and stated that "the utilization of saturation adjustment during diffusional growth introduces errors in CWC". Previous studies have identified limitations of using saturation adjustment, including overestimation of the condensate mass (Khain et al., 2015), and enhanced rain formation that reduces supercooled water in the mixed-phase region (Zhang et al., 2021). Previous studies that reported a decrease in liquid water content with the introduction of SIP processes (Phillips et al., 2022; Huang et al., 2024; Grzegorczyk et al., 2025) explicitly computed condensation and vapor deposition, unlike the LIMA scheme.

Another possible reason is that in the version of LIMA used in this study (v1.0; Vié et al., 2016), snow and graupel number concentrations are not prognostic. In contrast, the extended version of LIMA (v2.0; Taufour et al., 2024), includes prognostic number concentrations for all hydrometeor categories. Taufour et al. (2024) showed that in LIMA v1.0, snow and graupel form rapidly, consuming cloud droplets, raindrops, and ice crystals. In LIMA v2.0 their formation is more gradual.

A discussion about the insensitivity of CWC to SIP processes has been added in 4.4.2.

*"The weak sensitivity of CWC to SIP processes may result from several factors. Data sampling during the electrification period limits the detection of differences, which occur more significantly during the storm's mature stage. The use of saturation adjustment in LIMA, which enforces 100\% RH could be a constraint, as it can overestimate condensate mass and enhance rain formation, reducing supercooled water (Khain et al., 2014; Zhang, 2021). In contrast, studies showing stronger CWC responses to SIP explicitly compute condensation and vapor deposition (Phillips et al., 2022; Grzegorczyk et al., 2022; Huang et al., 2025). Additionally, in the version of LIMA used here (v1.0) snow and graupel number concentrations are not prognostic, potentially accelerating their formation and depleting liquid and small ice species as shown by Taufour et al. (2024) in comparisons with LIMA v2.0."*

3. I know that the simulations aren't meant to be compared to observations, but can the authors comment at least qualitatively on the magnitude of their results? Do previous observational studies support a nearly 10x increase in lightning flashes due to CCN?

We understand the desire to compare the present results with observations or to comment on them qualitatively. However, this is not an easy exercise. In terms of observational studies, most of them use aerosol optical depth (AOD) rather than aerosol number concentration and compare it with lightning discharges over large periods of time (yearly or per season data) (Shi et al., 2020; Proestakis et al., 2016; Dayeh et al., 2021; Wang et al., 2023; Altaratz et al., 2010). In general, the lightning activity is observed to increase between 50 and 150% when the aerosol loading increases (e.g., Naccarato et al., 2003 ; Thornton et al., 2017 ; Peterson, 2023). Studying the effect of aerosols on lightning activity over the Mediterranean sea, Proestakis et al. (2015) have found an enhancement factor of 9 of the number of lightning strikes between low and high AOD. From their Figure 7e, we can see that when the AOD increases from 0.1 to 0.5, the average number of lightning strikes increases between 30 and 280. In numerical studies, the lightning flash enhancement factor associated with different aerosol concentrations can reach higher values. Sun et al. (2023) simulated a multicellular storm during 6 hours, and found a 5-fold increase of the total lightning flashes when the CCN concentration was increased from 400 to 6400 $cm^{-3}$ (see their Figure 18).  Huang et al. (2025) showed an enhancement by a factor 60 when the CCN

number concentration is increased from 400 to 4000 cm$^{-3}$ (see their Figure 17) in a simulated squall line.

It must also be noted that the total number of flashes simulated by the model depends on the parameterization of the non-inductive process (Mansell et al., 2005 ; Barthe and Pinty, 2007a) and on the choice of the lightning flash parameters (Barthe and Pinty, 2007b). But, since the cloud electrification and lightning scheme parameters are held constant over the whole set of simulations, this should not affect the results.

A comparison of our results with the literature has been added in Section 3.3 of the new version of the manuscript:

*"These values are of the same order of magnitude as the ones in the literature. Sun et al. (2023) found that the total number of flashes was multiplied by 5 when $N_{CCN}$ increased from 400 to 6400 cm$^{-3}$ in a simulated multicell storm developing in a high CAPE environment. Huang et al. (2025) reported a nearly 60-fold increase of the total lightning number in a simulated squall line when the aerosol concentration increased from 400 to 4000 cm$^{-3}$. Observational studies based on AOD and lightning strikes data report similar increases in lightning activity, with enhancement factors ranging from 1.6 to 9 (Thornton et al., 2017; Naccarato et al., 2003; Proestakis et al., 2016)."*

**Minor Comments:**

1. There are several places where citations are needed, including Line 21-22, 48, and 69-72.

We added citations as recommended:
- Line 20: Reynolds et al. (1957) and Takahashi (1978)
- Lines 21-22: Norville et al. (1991) and Heldson et al. (2001)
- Line 48: van der Heever et al. (2006), Rosenfeld et al. (2008), and Sun et al. (2021)
- Lines 69-72: Mansell and Ziegler (2013), Tan et al. (2017), Yang et al. (2020), Sun et al. (2021), and Yang et al. (2024)

2. Line 86 – sentence is unfinished

The "…" marks the end of the sentence.

3. Lines 145-150 – how was MID-WARM triggered

The WARM case was triggered by a warm bubble of 1.5°C. In the first version of the manuscript, the description of the WARM and MID-WARM set-up was mixed. This is clarified in the new version of the manuscript.

4. Lines 155-161 – what size particles? Can more information be provided about the INP populations?

A single mode of CCN particles was set with a mean radius of 125 nm and a standard deviation (sigma) of 0.69. Concerning INP, a single mode was used with a mean radius of 0.8 μm and a sigma of 1.9. As in Vié et al. (2016) and as recommended by Phillips et al. (2008), the INP mode is composed of 61% of dust, 33% of black carbon and 6% of organic matter. We added these informations in the new version of the manuscript;

5. Just in general, the model setup information was minimal and could be described in greater detail, especially since model initialization files are not provided as part of the code/data/software availability.

In the new version of the manuscript, additional information about the model setup is given in Sections 2.2 and 2.3. The soundings of each idealized case are added as a supplementary material.

6. Line 193 – Not a complete sentence.

This sentence has been modified.

7. Line 238 – what does it mean that the simulations were treated together? That they were averaged together? Or that a representative simulation is shown?

It means that only one of the three simulations is shown. Only the simulation using $N_{INP}$ = 10 $L^{-1}$ is used in the rest of the article. This is clarified in the new version of the manuscript.

8. Line 248 – is the charging rate meant to have a unit of kC/s rather than just kC?

You are right: we checked our script and changed it to C/s in the new version of the manuscript.

9. Line 272 – by my eye CWC drops below 0.01 g/m^3 nearer to -20C than -10C.

It is corrected in the new version of the article.

10. Line 281 – effect of varying NCCN on what is mainly the same?

This is the effect of an increase of $N_{CCN}$ on cloud water content. It has been clarified in the text.

11. The Bergeron process is mentioned a few times. Typically I take this to mean the growth of ice and evaporation of droplets. But in a strong updraft, I assume that supersaturation is produced rapidly enough that supersaturation can be maintained with respect to both liquid and ice such that there is no Bergeron effect.

We agree that in convective updrafts supersaturation is usually maintained (Khain et al., 2012), and that the Bergeron effect operates under limited conditions, not consistently impacting ice particles and liquid droplets in mixed-phase clouds  (Korolev et al., 2007). In LIMA, the Bergeron process is not explicitly computed and the effect of riming of cloud droplets on aggregates and graupel is more significant. Therefore, we decided to remove in the manuscript the Bergeron effect.

12. Figure 10 – what are SIP tendencies exactly? The ice number production rate?

Yes, SIP tendencies correspond to the ice number production rate of each SIP process. It has been clarified in the legend of Figure 10.

13. It would be helpful to label the temperature lines in many of the figures.

You are right. The temperature lines have been labelled in the new version of the manuscript.

→ Overall, I found the manuscript to be overly detailed and a little tedious to read. I think that the main points could be conveyed with more concise text. But I leave this to the authors to decide.

In the new version of the manuscript, we've tried to make the text a little more concise. Given the large number of simulations to be processed, however, it is difficult to reduce the descriptions significantly. We have also modified the organization of the first part of the results following the recommendation of reviewer 2

[revised manuscript text omitted]

---

## Author Comment (AC2)

**Reply on RC2**

**RC2:** The manuscript "Distinct effects of several ice production processes on thunderstorm electrification and lightning activity" simulates three idealized storms, specified by the cloud base temperature and depth of the warm-phase layer, to assess the influence of aerosol concentration (CCN and INP) and three SIP processes (Hallett-Mossop rime splintering, raindrop shattering by freezing, and collision ice breakup) on cloud microphysics (ice crystal number concentration, cloud water content, graupel mass) and on electrification/lightning activity (charging rate on graupel, total number of flashes, and time of the first flash). The main results include an increase in lighting activity with the increase of CCN concentration up to a threshold as found in previous studies, but here it is shown that this threshold value varied depending on the INP concentration and type of storm. Each SIP process impacted the cloud electrification and lighting activity differently depending on the thickness of the cloud's warm-phase. The results also highlight that activating SIP processes in the simulations impacted more dramatically the lightning activity than varying/adjusting aerosol concentrations (CCN or INP). In general, the study is well-structured and presents valuable and relevant contributions within the scope of ACP, but there are some inconsistencies mainly in sections 3 and 4. My comments are included below.

We thank the reviewers for their time and efforts in reviewing our manuscript. The responses to their comments are addressed below.

**General Comments:**

● Aerosol concentration and SIP process for control run In line 176, it is mentioned that the aerosol concentrations were kept constant at N_CCN = 1000 cm^-3 and N_INP = 10 L^-1 when analyzing the impact of the SIP processes. Why were these values chosen? Was there an additional evaluation to arrive at these values? Was this choice made based on a paper? If so, I recommend including the citation. CCN concentration of 1000 cm^-3 could be considered part of the high range of CCN concentration (Mansell and Ziegler, 2013). If not, I suggest including, mentioning or highlighting what would be realistic values or range of values for the three types of storms, since the chosen sensitivity range spans a more extensive range not explored by other studies as mentioned in lines 165-174. Additionally, in line 174, why only the HM process is activated in the first set of simulation varying aerosol concentrations? Could the reasoning for this decision also be included?

As pointed out by the reviewers, we have not explained the choice of CCN and INP concentrations for simulations studying sensitivity to SIPs. As stated in the manuscript *"aerosol concentrations are kept constant with $N_{CCN}$ = 1000 cm$^{-3}$ and $N_{INP}$ = 10 L$^{-1}$"*. These concentrations are supposed to be representative of average aerosol conditions. Rose et al. (2021) have surveyed aerosol concentrations using the network of Global Atmosphere Watch (GAW) near-surface observatories. Over the continents, total aerosol concentrations range between 1000 cm$^{-3}$ in rural areas to $10^4$ cm$^{-3}$ in urban areas. Using particle number concentration in the range 100-500 nm as a proxy for potential CCN population (see their Figure 12), they showed that, from this dataset, the potential CCN concentration ranges between a few hundreds to a few thousands particles cm$^{-3}$. Moreover, in their modeling

study, Mansell and Ziegler (2013) used 13 base values of CCN concentration (50, 100, 200, 300, 500, 700, 1000, 1500, 2000, 3000, 4000, 5000, and 8000 $cm^{-3}$) for which 1000 $cm^{-3}$ is the median value. Sun et al. (2021) used an initial CCN concentration of 1200 $cm^{-3}$, typical of continental values. Regarding INP concentrations, Figure 1-10 of Kanji et al. (2017) combines various measures of INP concentrations at different temperatures. At temperatures colder than -15°C, most studies exhibit INP concentrations between 0.5 and 50 $L^{-1}$. Phillips et al. (2007) performed sensitivity tests on $N_{INP}$ in a convective case. Observations showed $N_{INP}$ of around 3 $L^{-1}$, and two tests representing high and extreme $N_{INP}$ cases were simulated with $N_{INP}$ of 30 and 3000 $L^{-1}$, respectively. We are aware that the three storms might have different aerosol conditions. However we decided to set the same initial aerosol concentrations to make it easier to compare the activity of the SIP mechanisms in each storm.

In order to justify our choice of $N_{CCN}$ and $N_{INP}$ for the sensitivity tests dedicated to the SIP mechanisms, a paragraph has been added in Section 2.3:

"*In this series of simulations, aerosol concentrations representative of average aerosol conditions are used. Rose et al. (2021) have surveyed aerosol concentrations using the network of Global Atmosphere Watch (GAW) stations. Using particle number concentration in the range 100-500 nm as a proxy for potential CCN population, they showed that the potential CCN concentration ranges between a few hundreds to a few thousands particles $cm^{-3}$ over the continents. Mansell and Ziegler (2013) and Sun et al. (2021) used values around 1000 $cm^{-3}$ in their modeling studies. Regarding INP concentrations, Kanji et al. (2017) showed that most studies exhibit INP concentrations between 0.5 and 50 $L^{-1}$ at temperatures colder than -15°C. Therefore, $N_{CCN}$ = 1000 $cm^{-3}$ and $N_{INP}$ = 10 $L^{-1}$ are used in all these simulations.*"

For decades, 2-moment schemes include a parameterization of the Hallett-Mossop ice multiplication mechanism (e.g., Ferrier, 1994 ; Meyers et al., 1997 ; Seifert and Beheng, 2006 ; Vié et al., 2016). On the contrary, the collisional ice breakup (CIBU) and the raindrop shattering by freezing (RDSF) mechanisms have been only recently included in microphysics schemes (Phillips et al., 2017; Hoarau et al., 2018, Phillips et al., 2018; Sullivan et al, 2018). In particular, uncertainties remain regarding the number of fragments produced by these processes (Grzegorczyk et al., 2025). Moreover, today, CIBU and RDSF can be activated or deactivated in LIMA at the user's discretion while HM is systematically activated (Taufour et al., 2024). For these two reasons, it was decided to keep HM active in the first series of simulations. A short statement has been added in Section 2.3 to justify it:

"*In this first set of simulations, only the HM process as a SIP mechanism is activated. For decades, two-moment schemes include a parameterization of the HM process (e.g. Ferrier, 1994; Straka and Mansell, 2005; Seifert and Beheng, 2006; Vié et al., 2016), while the CIBU and RDSF mechanisms have been only recently included in microphysics schemes (Phillips et al., 2017a, 2018; Hoarau et al., 2018; Sullivan et al., 2018; Grzegorczyk et al., 2025a) with uncertainties remaining regarding the number of fragments produced by these processes (Grzegorczyk et al., 2025b). Moreover, CIBU and RDSF can be activated or deactivated in LIMA at the user's discretion while HM is systematically activated. Therefore, it was decided to keep HM active in these first series of simulations.*"

● Charge separation parameterization In line 163, the authors state that "... the non-inductive charge separation is parameterized following Takahashi (1978)..." Although it is mentioned in line 528 "... there is still no consensus on the parameterization of the non-inductive process, and several existing parameterizations should be tested." I expected the manuscript to provide more detail on the implementation of this parameterization and to discuss the potential implications its selection may have on the results. This is particularly important given that the Saunders and Peck (1998) scheme is widely used and has been shown to also successfully reproduce inverted-polarity charge structures, as demonstrated by for example by Kuhlman et al. (2006).

Several parameterizations of the non-inductive mechanism (e.g., Takahashi, 1978 ; Saunders et al., 1991 ; Saunders and Peck, 1998) are available in Meso-NH as described in Barthe and Pinty (2007) and Tsenova et al. (2013). We are aware that the Saunders and Peck's (1998) formulation has been frequently used in many numerical studies and was able to reproduce charge structures observed in storms (Kuhlman et al., 2006 ; Fierro et al., 2006, 2013 ; Mansell et al., 2010 ; Sun et al., 2021). However, we opted for the parameterization of Takahashi (1978) which was also used in various numerical studies (Barthe et al., 2007 ; Barth et al., 2007 ; Pinty et al., 2013 ; Bovalo et al., 2019 ; Popova et al., 2022 ; Phillips et al., 2022). This choice was motivated by recent laboratory studies that show strong similarities between the charge reversal line in Takahashi (1978) and the ones in Pereyra et al. (2000), Saunders et al. (2006) or Emersic and Saunders (2010). Some discrepancies appear when the temperature and the liquid water content decrease. The recent laboratory experiment by Luque et al. (2020) has found similar behavior as Pereyra et al. (2000), Saunders et al. (2006), and Emersic and Saunders (2010), meaning that the parameterization of Takahashi (1978) in Meso-NH should be modified in the future.
Numerous studies have shown that the choice of the non inductive process parameterization can strongly influence model results, both in terms of charge structure and number of flashes (Helsdon et al., 2001 ; Altaratz et al., 2005 ; Mansell et al., 2005 ; Barthe and Pinty, 2007 ; Fierro et al., 2006 ; Kuhlman et al., 2006 ; Tsenova et al., 2013). Therefore, the charge structures shown in this study would be different if the Saunders and Peck (1998) parameterization was used. However, the objective of this study is not to evaluate which parameterization of the non-inductive process is the best suited for storm modeling, but rather to isolate and explore the effect of ice production on cloud electrification. An evaluation of the non-inductive mechanism parameterization should be done in a real case simulation with microphysical, dynamical and electrical observations.

In the new version of the manuscript, we added a paragraph in Section 2.2 to justify the choice of the non-inductive process parameterization, and another one in Sections 3.3 to discuss the uncertainties about the parameterization of the non-inductive process.

*"The choice of the non inductive charging parameterization can impact model results, both in terms of charge structure and total number of flashes (Helsdon Jr. et al., 2001; Altaratz et al., 2005; Mansell et al., 2005; Barthe and Pinty, 2007a; Fierro et al., 2006; Kuhlman et al., 2006; Tsenova et al., 2013). Both the parameterizations of Saunders and Peck (1998) and Takahashi (1978) have been widely used to simulate the electrical activity of thunderstorms. However, recent laboratory studies have shown strong similarities between the charge reversal line in Takahashi (1978) and the ones in Pereyra et al. (2000), Saunders et al. (2006) or Emersic and Saunders (2010), leading us to choose the parameterization of Takahashi (1978) for the non-inductive charge separation in this study."*

*"Variations in aerosol concentrations modify both the amplitude and the sign of the charge exchanged during the non-inductive process, and thus the polarity of the cloud's charge structures. Numerous studies have shown that the choice of the non-inductive process parameterization can strongly influence model results, both in terms of charge structure and number of flashes (Helsdon Jr. et al., 2001; Altaratz et al., 2005; Mansell et al., 2005; Barthe and Pinty, 2007a; Fierro et al., 2006; Kuhlman et al., 2006; Tsenova et al., 2013). Therefore, the charge structures shown in this study would be different if the Saunders and Peck (1998) parameterization was used. However, the objective of this study is not to evaluate which parameterization of the non-inductive process is the best suited for storm modeling, but rather to isolate and explore the effect of ice production on cloud electrification."*

● Charge density instead of just the charging rate on graupel Figures 3 and 9 only present the charging rate on graupel, but this information alone does not provide a clear indication of the storm's overall charge structure. I would strongly suggest providing cross-sectional plots of the charge density to reduce ambiguity in the interpretation and validation of the results. Additionally, Figure 1 presents only the thickness of the warm, mixed and cold-phase regions of the three idealized storms. It would be beneficial to include additional context of the simulation results such as plots of the simulated radar reflectivity to illustrate how these storms evolve and to better connect to the idealized setups.

A general problem when carrying out a large series of numerical simulations is to illustrate the results in a synthetic way. Clearly, it is not possible to produce vertical cross-sections for all storms. Changes in aerosol concentrations, and the activation or non-activation of SIPs modify the cloud structure, especially at the end of the simulation.

However, we understand the need to have more information about the context of each idealized storm. So as not to make the already relatively long article too long, we have decided to include both the soundings used to initialize the idealized storms (Figure S1), and representative vertical cross-sections of the three simulated storms (Figure S2) as Supplementary Material. The vertical cross-sections have been plotted for the 3 storms, but for only one simulation each. We have selected to show cross-sections for the following set up: $N_{CCN}$ = 1000 cm$^{-3}$, $N_{INP}$ = 10 L$^{-1}$, HM activated, CIBU and RDSF deactivated. This setup corresponds to the common simulation between the two series of sensitivity tests.

In the same way, it is not an easy task to produce synthetic and representative plots of the charge structure for the 87 simulations. The large number of simulations is compounded by the complexity of the electric charge structure. As shown in Figure 1 below, and as stated in Stolzenburg et al. (1998), the charge structure may differ depending on the region of the convective system. But, as mentioned by the reviewers, knowledge of the electrical charge structure is an important piece of information. We have therefore decided to include a new figure showing charge density. As it is impossible to show 87 vertical cross-sections of the charge structure, we decided to plot the average of the positive and negative charge density for each simulation. Since the electric charge layers are not horizontally aligned, the resulting profiles are quite noisy, but general trends are visible. Due to the difficulty of assigning a distinct charge structure from the average of positive and negative charges, it was decided to include this figure as Supplementary Material (Figures S3 and S4).

[Figure]

*Figure 1: vertical cross-section of the total charge density (nC m⁻³) for the WARM, MID-WARM and COLD simulations at 30 min. The cross-sections are shown for the simulation with $N_{CCN}$ = 1000 cm⁻³, $N_{INP}$ = 10 $L^{-1}$, and HM activated. The vertical cross-section passes through the maximum updraft.*

● Comparison with Phillips and Patade (2022) The results for the cold case are compared with those of Phillips and Patade (2022), showing consistency on the importance of the CIBU process, as noted in line 516: "This is consistent with Phillips and Patade (2022) results for a cold-base thunderstorm in which HM and RDSF are almost inactive." There are more details in the introduction from Phillips and Patade (2022) and the effect of CIBU on CWC in line 66 "Phillips and Patade (2022) found that the most active SIP process was breakup during ice-ice collisions. This process, acting as a sink of liquid water content, has the ability to alter the polarity of the charge Graupel acquires and, consequently, the electric charge structure." However, in line 437 and referring to figure 12 the manuscript states that: "The COLD case does not show any impact of the SIP processes on the average CWC profile in the early cloud electrification stage... As cloud electrification starts during the development stage of the cloud, SIP processes have not yet consumed CWC." The comparison as currently presented appears to lack consistency. I recommend revising the text and revisiting the simulation/analysis to address potential contradictions of the results also within the manuscript and ensure a clearer discussion.

We agree that the comparison with Phillips and Patade (2022) lacks consistency. The insensitivity of CWC to SIP processes in our study was also pointed out by the RC1. Several factors mostly in LIMA could explain why in our simulations CWC shows little changes despite the production of a high number of ice crystals by SIP processes.

Firstly, it partly results from the data being sampled during the electrification period defined in this study. At the mature stage, the simulations start to diverge; the ALLSIP simulation having the least CWC. However the differences remain small compared to the stronger sensitivity of CWC to aerosol concentrations.

A possible explanation for this weak response is the use of a saturation adjustment scheme in LIMA. This adjustment, applied after all other microphysical processes, forces the environment to reach a strict equilibrium at water saturation at the end of each time step, by condensing water vapor or evaporating cloud droplets depending on whether the air is supersaturated or sub-saturated. In a general way, Khain et al. (2015) examined the bin and bulk parameterizations of microphysics and stated that "the utilization of saturation adjustment during diffusional growth introduces errors in CWC". Previous studies have identified limitations of using saturation adjustment, including overestimation of the condensate mass (Khain et al., 2015), and enhanced rain formation that reduces supercooled water in the mixed-phase region (Zhang et al., 2021). Previous studies that reported a decrease in liquid water content with the introduction of SIP processes (Phillips et al., 2022; Huang et al., 2024; Grzegorczyk et al., 2025) explicitly computed condensation and vapor deposition, unlike the LIMA scheme.

Another possible reason is that in the version of LIMA used in this study (v1.0; Vié et al., 2016), snow and graupel number concentrations are not prognostic. In contrast, the extended version of LIMA (v2.0; Taufour et al., 2024), includes prognostic number concentrations for all hydrometeor categories. Taufour et al. (2024) showed that in LIMA v1.0, snow and graupel form rapidly, consuming cloud droplets, raindrops, and ice crystals. In LIMA v2.0 their formation is more gradual.

A discussion was added about the insensitivity of CWC to SIP processes and contradiction with Phillips and Patade (2022) results in section 4.2.2.:

*"The weak sensitivity of CWC to SIP processes may result from several factors. Data sampling during the electrification period limits the detection of differences, which occur more significantly during the storm's mature stage. The use of saturation adjustment in LIMA, which enforces 100\% RH could be a constraint, as it can overestimate condensate mass and enhance rain formation, reducing supercooled water (Khain et al., 2015; Zhang, 2021). In contrast, studies showing stronger CWC responses to SIP explicitly compute condensation and vapor deposition (Phillips et al., 2022; Grzegorczyk et al., 2022; Huang et al., 2025). Additionally, in the version of LIMA used here (v1.0) snow and graupel number concentrations are not prognostic, potentially accelerating their formation and depleting liquid and small ice species as shown by Taufour et al. (2024) in comparisons with LIMA v2.0."*

**Specific Comments:**

**Abstract**

Line 13: What impact on electrification is this referring to? Is it regarding the polarity, the charge magnitude, number of flashes, …?

This part of the sentence was not clear and has been modified.

**Introduction**

Suggest include citations for the sentences starting in lines 20 and 21.

We added citations as recommended:

- Line 20: Reynolds et al. (1957) and Takahashi (1978)

- Lines 21-22: Norville et al. (1991) and Heldson et al. (2001)

**2.1.1 Microphysical scheme**

In lines 101 and 105, the authors introduce abbreviations for the SIP processes: collisional ice break-up as CIBU and raindrop shattering freezing as RDSF. But in line 99, there is no mention of the abbreviation of the Hallett-Mossop process as HM. Additionally to maintain consistency, in line 315 and 505, this process is referred to as rime splintering, when throughout the manuscript HM process has been used. This term could be introduced in line 99 as well.

We added the mention of the abbreviation HM and introduced the term "rime splintering".

In lines 108-118, the manuscript provides implemented equations, expressions and values for the RDSF process. But the same treatment is not given to the other SIP processes HM and CIBU. Is there a reason for expanding the explanation just for RDSF and not the other processes? Was the RDSF implementation different from the cited studies?

We provided a more detailed description of the RDSF process as it is the first time this process is activated in a study with the atmospheric model Meso-NH and its microphysics scheme LIMA. We modified the text in Section 2.1.1 to make clearer that the HM and CIBU parameterizations in LIMA were already presented in Vié et al. (2016) and Hoarau et al. (2018), respectively.

The units for INP concentrations are given in $L^1$, but in line 172, a reference from concentrations used in another study are given in $cm^{-3}$. Writing the concentrations in the same units would help the reader to compare the range and values considered.

The INP concentrations used in Yang et al. (2000) have been converted in $L^{-1}$.

**Results: Sections 3 and 4**

Recommend maintaining a structure in the results sections 3 and 4. In section 3, it is presented the following subsections:
    3 Aerosol impact on cloud electrification and lightning activity
    3.1 Electrical activity
    3.2 Microphysical structure of the storms
    3.2.1 Cloud water content
    3.2.2 Ice crystal concentration
    3.2.3 Graupel mass
    3.3 The relationship between aerosols, microphysics and electrification
In section 4, they are:
    4 Effect of secondary ice production on cloud electrification and lightning activity
    4.1 Electrical activity
    4.2 Microphysics
    4.2.1 Ice crystal number concentration
    4.2.2 Cloud water content

4.2.3 Graupel mass
4.2.4 The relationship between SIP processes, microphysics and electrification

So, the subsection titles and the order they appeared are modified from what was in section 3. Recommend keeping this consistent.

We modified the subsections in the new version of the manuscript. However, we decided to keep the structure of Section 4, and to implement it in Section 3. Since this paper deals with ice production processes, it seems logical to first look at ice crystal number concentration.

In line 190: "... we will focus on the modification of the electrical activity and of the microphysics of each idealized case due to the sensitivity tests rather than on the differences between the three cases with the same aerosol concentration and SIP process conditions." But, in line 373 the results are compared across storms under the same set of conditions: "This enhancement is 7 times higher in the WARM case than in the MID-WARM case. " How much are their respective increases compared to just HM or HM+CIBU?

You are right. This sentence has been removed in the new version of the manuscript, and the previous one has been modified to give more information on the enhancement factor for each simulation.

In line 268, when referring to the Takahashi diagram, I would suggest citing the paper, since there are a couple of Takahashi's papers in the References section.

The citation to Takahashi (1978) has been added.

There are several mentions of high and low values for N_CCN and N_INP but the range is only specified later in the section. I would suggest making it more clear at the beginning of the section or on the sensitivity test section the ranges for low, medium and high N_CCN and N_INP.

The ranges for low,  medium and high $N_{CCN}$ and $N_{INP}$ are now specified in Section 2.3:

"*In the remainder of the paper, low $N_{CCN}$ refers to 500 cm$^{-3}$, medium to  1000 and 5000 cm$^{-3}$ and high to 8000 and 10,000 cm$^{-3}$. Low $N_{INP}$ corresponds to 0.1, 1 and 10 L$^{-1}$, medium to 100 L$^{-1}$ and high to 1000 L$^{-1}$.*"

For the warm case, what is the range that the HM process is maximum/most intense, since the following sentences seem to disagree? In line 315: "That is why the HM process is the most intense for intermediate values of N_CCN in the WARM and MID-WARM cases." But in line 309: "For the WARM and COLD cases, the HM process rate is maximum for high N_INP (≥ 100 L^−1) and high N_CCN (≥ 5000 cm^−3)."

At line 315, we wanted to underline the threshold effect by using the expression "intermediate values of $N_{CCN}$". To remain consistent with the definition of low, medium and high $N_{CCN}$ in Section 2.3, we change it to : "*not for the highest $N_{CCN}$ but at 8000 cm$^{-3}$ and 1000 cm$^{-3}$ in the WARM and MID-WARM cases, respectively*".

At line 309, the text was also changed to: "*...is maximum for $N_{INP}$ ≥ 100 L$^{-1}$ and $N_{CCN}$ ≥ 5000 cm$^{-3}$*".

Line 325: "It suggests graupel mass is not a limiting ingredient for cloud electrification, but it can modulate the amplitude of the charge exchanged during the non-inductive process." I would recommend explaining this better as it is not clear to me the results are suggesting this.

We removed this sentence as it does not apply for all cases. High graupel mass is correlated with high non inductive charging rate and total flash number especially in the MID-WARM case, but in the two other cases the relationship between graupel mass and electric activity is not clear.

There are numerous instances where the word "whatever" is used. I would recommend replacing it with "regardless of" or "independent of".

All whatever occurrences were replaced with "regardless of", "independent of", "for any values of" and "across all …".

Line 338: "The formation is accelerated but the intensity is weaker leading to a lower graupel mass at high N_INP." The intensity of what is being referenced here?

We talk about the intensity of graupel mass growth. This is specified in the new version of the manuscript.

Lines 378-380. These sentences could be combined to avoid repetition.

The two sentences have been combined: "*In the WARM and MID-WARM cases, the dramatic increase in total flashes is largely due to the combined and significant impact of the RDSF and CIBU processes.*"

Line 395: "In the WARM case, the HM process tendency is identical for the two pairs of simulations HM and HM+CIBU (6.5 x 10^9 kg^−1 s^−1), and HM+RDSF and ALLSIP (7.1 and 7.2 x 10^9 kg^−1 s^−1)..." 7.1 and 7.2 are not identical values.

We changed the word "identical" with "similar".

Is the result in line 397: "The CIBU process is very efficient in producing ice crystals over the whole mixed and cold cloud depth, leading to an increase of ice crystal number concentration by around two orders of magnitude (green and blue lines in Fig. 11a)." in comparison to NOSIP or HM simulation?

It is in comparison to the NOSIP simulation (blue line). It is clarified in the new version of the manuscript.

Line 399: "RDSF is the most efficient SIP in this storm; it induces a maximum of 1000 L^−1 (orange line in Fig. 11a)." What altitude and/or temperature does this correspond to?

We added "at 15 km altitude".

Line 400: "Despite being the most active at -15degC, the RDSF process results in high N_i throughout the whole mixed and cold cloud depth..." There is not an isotherm line for -15 degC, so what altitude does it correspond to?

It corresponds to 7.5 km altitude ; it was added in the manuscript.

Line 402: "When the three SIP processes are active (ALLSIP), they add up to produce mean ice crystal number concentration that reaches a maximum of 1500 L^−1." Is this maximum ice crystal concentration at the same altitude of the 1000 L^_1 peak for process RDSF (line 399)?

The peak in ALLSIP is at the same altitude as in the RDSF simulation.

Is "the HM+RDSF simulation presents lower values of ice crystal concentration along the vertical profile" in line 410 a comparison to the lower values in the HM+CIBU simulations?

It is in comparison to the overall vertical profile of HM+CIBU simulation. Although a similar peak at an altitude of 8 km, the HM+CIBU simulation presents a higher ice crystal concentration at other altitudes.

Line 413: "Actually, RDSF needs a deep warm-phase cloud depth and a moderate updraft which will help raindrops to grow and to be lifted up to the right temperature region (Sullivan et al., 2018)" what is the right temperature region?

The right temperature region is around -15 °C where the maximum probability of shattering is reached in the RDSF parameterization. This is now specified in the text.

Line 414: "Interestingly, in the ALLSIP simulation, the RDSF process 415 tendency is tripled compared to the HM+RDSF simulation." This refers to figure 10b, right? Add it here.

You are right. The reference to this figure has been added.

Line 421: They increase the mean ice crystal number concentration by up to a factor of 1000 in the temperature range in which they are active." What is this temperature range?

HM process is active between -8 and -3°C, and CIBU is active in the mixed and cold phase region. It has been specified in the text.

Line 428: "In the MID-WARM case, CWC is higher in the NOSIP simulation than in all simulations where SIP processes are activated near the 0degC isotherm." It looks like it is activated until close to -10degC isotherm.

You are right, this sentence is not clear. It has been modified in the revised manuscript: "*In the MID-WARM case, in the altitude range between the 10°C and -10°C isotherms, CWC is higher in the NOSIP simulation than in all simulations where SIP processes are activated.*"

Line 439: "... the non-inductive charging process only occurs at high altitude (between 7.5 and 11 km), where ice crystals are available…" Figure 9 shows charge separation occurring for ALLSIP simulation from 5 km altitude.

We forgot to specify that this comment was only for the NOSIP simulation; it is now specified.

In line 473, what does "different cloud electrification onsets" mean? Do the plots in Figure 4 for N_CCN = 1000 cm^-3 (black line), N_INP = 10 L^_1 (third row) match the ones for Figure 12 with HM process (black line)? Could this correspondence of control cases be included in the manuscript?

Cloud electrification onsets refer to the beginning of cloud electrification defined at the beginning of section 3.2.

The black lines in Figures 9 and 12 are from the same simulation. We added a sentence about this correspondence in Section 2.3.

**Conclusions**

Line 491: "As for the HM process, it is maximum at intermediate or high N_CCN levels (1000 - 10,000 cm^−3)" Is maximum at producing ice crystals? Also, inconsistent definition for the high range of N_CCN in line 308. Lines 497-500. Combine sentences to avoid repetition.

In the revised version of the article, we only kept the range of values for $N_{CCN}$ without specifying the intensity of $N_{CCN}$ levels.

The two sentences at lines 497-500 have been combined.

Line 500: "Mansell and Ziegler (2013) attributed the decrease of lightning activity with NCCN to the HM process…" A decrease is observed once the threshold value is exceeded?

Yes, the total number of flashes reaches its maximum at the $N_{CCN}$ threshold value and then decreases at higher $N_{CCN}$ (see their figure 9 for the HM1 simulation).

Line 503: "Thus, both particles has to be taken into account to ..." Adjust to: Both aerosol particles have to be …

Done.

**References**

Altaratz, O., Reisin, T., and Levin, Z.: Simulation of the electrification of winter thunderclouds using the three-dimensional Regional Atmospheric Modeling System (RAMS) model: Single cloud simulations, J. Geophys. Res.: Atmos., 110, 2004JD005 616, https://doi.org/10.1029/2004JD005616, 2005.

Barth, M. C., Kim, S.-W., Wang, C., Pickering, K. E., Ott, L. E., Stenchikov, G., Leriche, M., Cautenet, S., Pinty, J.-P., Barthe, C., Mari, C., Helsdon, J. H., Farley, R. D., Fridlind, A. M., Ackerman, A. S., Spiridonov, V., and Telenta, B.: Cloud-scale model intercomparison of chemical constituent transport in deep convection, Atmos. Chem. Phys., 7, 4709–4731, https://doi.org/10.5194/acp-7-4709-2007, 2007.

Barthe, C. and Pinty, J.: Simulation of electrified storms with comparison of the charge structure and lightning efficiency, J. Geophys. Res. Atmos., 112, 2006JD008 241, https://doi.org/10.1029/2006JD008241, 2007.

Bovalo, C., Barthe, C., and Pinty, J.: Examining relationships between cloud-resolving model parameters and total flash rates to generate lightning density maps, Q. J. Roy. Meteor. Soc., 145, 1250–1266, https://doi.org/10.1002/qj.3494, 2019.

Emersic, C. and Saunders, C.: Further laboratory investigations into the Relative Diffusional Growth Rate theory of thunderstorm electrification, Atmos. Res., 98, 327–340, https://doi.org/10.1016/j.atmosres.2010.07.011, 2010.

Ferrier, B. S.: A Double-Moment Multiple-Phase Four-Class Bulk Ice Scheme. Part I: Description, J. Atmos. Sci., 51, 249 – 280, https://doi.org/10.1175/1520-0469(1994)051<0249:ADMMPF>2.0.CO;2, 1994.

Fierro, A. O., M. S. Gilmore, E. R. Mansell, L. J. Wicker, and J. M. Straka: Electrification and Lightning in an Idealized Boundary-Crossing Supercell Simulation of 2 June 1995. Mon. Wea. Rev., 134, 3149–3172, https://doi.org/10.1175/MWR3231.1, 2006.

Fierro, A. O., E. R. Mansell, D. R. MacGorman, and C. L. Ziegler: The Implementation of an Explicit Charging and Discharge Lightning Scheme within the WRF-ARW Model: Benchmark Simulations of a Continental Squall Line, a Tropical Cyclone, and a Winter Storm. Mon. Wea. Rev., 141, 2390–2415, https://doi.org/10.1175/MWR-D-12-00278.1., 2013.

Grzegorczyk, P., Wobrock, W., Canzi, A., Niquet, L., Tridon, F., and Planche, C.: Investigating secondary ice production in a deep convective cloud with a 3D bin microphysics model: Part II - Effects on the cloud formation and development, Atmos. Res., 314, 107 797, https://doi.org/10.1016/j.atmosres.2024.107797, 2025.

Grzegorczyk, P., Wobrock, W., Canzi, A., Niquet, L., Tridon, F., and Planche, C.: Investigating secondary ice production in a deep convective cloud with a 3D bin microphysics model: Part I - Sensitivity study of microphysical processes representations, Atmos. Res., 313, 107 774, https://doi.org/https://doi.org/10.1016/j.atmosres.2024.107774, 2025.

Helsdon Jr., J. H., Wojcik, W. A., and Farley, R. D.: An examination of thunderstorm-charging mechanisms using a two-dimensional storm electrification model, J. Geophys. Res.: Atmos., 106, 1165–1192, https://doi.org/10.1029/2000JD900532, 2001.

Hoarau, T., Pinty, J.-P., and Barthe, C.: A representation of the collisional ice break-up process in the two-moment microphysics LIMA v1.0 scheme of Meso-NH, Geosci. Model Dev., 11, 4269–4289, https://doi.org/10.5194/gmd-11-4269-2018, 2018.

Kanji, Z. A., L. A. Ladino, H. Wex, Y. Boose, M. Burkert-Kohn, D. J. Cziczo, and M. Krämer: Overview of Ice Nucleating Particles. *Meteor. Monogr.*, 58, 1.1–1.33, https://doi.org/10.1175/AMSMONOGRAPHS-D-16-0006.1, 2017.

Kuhlman, K. M., C. L. Ziegler, E. R. Mansell, D. R. MacGorman, and J. M. Straka: Numerically Simulated Electrification and Lightning of the 29 June 2000 STEPS Supercell Storm. *Mon. Wea. Rev.*, 134, 2734–2757, https://doi.org/10.1175/MWR3217.1, 2006

Luque, M. Y., Nollas, F., Pereyra, R. G., Bürgesser, R. E., & Ávila, E. E.: Charge separation in collisions between ice crystals and a spherical simulated graupel of centimeter size. J. Geophys. Res. Atmos., 125, e2019JD030941. https://doi.org/10.1029/2019JD030941, 2020

Mansell, E. R., MacGorman, D. R., Ziegler, C. L., and Straka, J. M.: Charge structure and lightning sensitivity in a simulated multicell thunderstorm, J. Geophys. Res.: Atmos., 110, 2004JD005 287, https://doi.org/10.1029/2004JD005287, 2005.

Mansell, E. R., C. L. Ziegler, and E. C. Bruning: Simulated Electrification of a Small Thunderstorm with Two-Moment Bulk Microphysics. *J. Atmos. Sci.*, **67**, 171–194, https://doi.org/10.1175/2009JAS2965.1, 2010.

Mansell, E. R. and Ziegler, C. L.: Aerosol Effects on Simulated Storm Electrification and Precipitation in a Two-Moment Bulk Microphysics Model, J. Atmos. Sci., 70, 2032–2050, https://doi.org/10.1175/JAS-D-12-0264.1, 2013.

Meyers, M. P., Walko, R. L., Harrington, J. Y., and Cotton, W. R.: New RAMS cloud microphysics parameterization. Part II: The two-moment scheme, Atmos. Res., 45, 3–39, https://doi.org/10.1016/S0169-8095(97)00018-5, 1997.

Norville, K., Baker, M., and Latham, J.: A numerical study of thunderstorm electrification: Model development and case study, J. Geophys. Res. Atmos., 96, 7463–7481, https://doi.org/https://doi.org/10.1029/90JD02577, 1991.

Pereyra, R. G., Avila, E. E., Castellano, N. E., and Saunders, C. P. R.: A laboratory study of graupel charging, J. Geophys. Res. Atmos., 105, 20 803–20 812, https://doi.org/10.1029/2000JD900244, 2000.

Phillips, V. T. J., L. J. Donner, and S. T. Garner: Nucleation Processes in Deep Convection Simulated by a Cloud-System-Resolving Model with Double-Moment Bulk Microphysics. J. Atmos. Sci., 64, 738–761, https://doi.org/10.1175/JAS3869.1, 2007.

Phillips, V. T. J., Yano, J.-I., Formenton, M., Ilotoviz, E., Kanawade, V., Kudzotsa, I., Sun, J., Bansemer, A., Detwiler, A. G., Khain, A., and Tessendorf, S. A.: Ice Multiplication by Breakup in Ice–Ice Collisions. Part II: Numerical Simulations, J. Atmos. Sci., 74, 2789–2811, https://doi.org/10.1175/JAS-D-16-0223.1, 2017.

Phillips, V. T. J., S. Patade, J. Gutierrez, and A. Bansemer: Secondary Ice Production by Fragmentation of Freezing Drops: Formulation and Theory. J. Atmos. Sci., 75, 3031–3070, https://doi.org/10.1175/JAS-D-17-0190.1, 2018.

Phillips, V. T. J. and Patade, S.: Multiple Environmental Influences on the Lightning of Cold-Based Continental Convection. Part II: Sensitivity Tests for Its Charge Structure and Land–Ocean Contrast, J. Atmos. Sci., 79, 263–300, https://doi.org/10.1175/JAS-D-20-0234.1, 2022.

Pinty, J.-P., Barthe, C., Defer, E., Richard, E., and Chong, M.: Explicit simulation of electrified clouds: From idealized to real case studies, Atmos. Res., 123, 82–92, https://doi.org/10.1016/j.atmosres.2012.04.008, 2013.

Popová, J., Sokol, Z., Šlegl, J., Wang, P., and Chou, Y.-L.: Research cloud electrification model in the Wisconsin dynamic/microphysical model 2: Charge structure in an idealized thunderstorm and its dependence on ion generation rate, Atmos. Res., 270, 106 090, https://doi.org/10.1016/j.atmosres.2022.106090, 2022.

Reynolds, S. E., Brook, M., and Gourley, M. F.: Thunderstorm charge separation, J. Atmos. Sci., 14, 426–436, https://doi.org/10.1175/1520-0469(1957)014<0426:TCS>2.0.CO;2, 1957.

Rose, C., Collaud Coen, M., Andrews, E., Lin, Y., Bossert, I., Lund Myhre, C., Tuch, T., Wiedensohler, A., Fiebig, M., Aalto, P., Alastuey, A.,Alonso-Blanco, E., Andrade, M., Artíñano, B., Arsov, T., Baltensperger, U., Bastian, S., Bath, O., Beukes, J. P., Brem, B. T., Bukowiecki, N., Casquero-Vera, J. A., Conil, S., Eleftheriadis, K., Favez, O., Flentje, H., Gini, M. I., Gómez-Moreno, F. J., Gysel-Beer, M., Hallar,A. G., Kalapov, I., Kalivitis, N., Kasper-Giebl, A., Keywood, M., Kim, J. E., Kim, S.-W., Kristensson, A., Kulmala, M., Lihavainen,H., Lin, N.-H., Lyamani, H., Marinoni, A., Martins Dos Santos, S., Mayol-Bracero, O. L., Meinhardt, F., Merkel, M., Metzger, J.-M., Mihalopoulos, N., Ondracek, J., Pandolfi, M., Pérez, N., Petäjä, T., Petit, J.-E., Picard, D., Pichon, J.-M., Pont, V., Putaud, J.-P., Reisen, F., Sellegri, K., Sharma, S., Schauer, G., Sheridan, P., Sherman, J. P., Schwerin, A., Sohmer, R., Sorribas, M., Sun, J., Tulet, P., Vakkari, V., Van Zyl, P. G., Velarde, F., Villani, P., Vratolis, S., Wagner, Z., Wang, S.-H., Weinhold, K., Weller, R., Yela, M., Zdimal, V., and Laj, P.: Seasonality of the particle number concentration and size distribution: a global analysis retrieved from the network of Global AtmosphereWatch (GAW) near-surface observatories, Atmos. Chem. Phys, 21, 17 185–17 223, https://doi.org/10.5194/acp-21-17185-2021, 2021.

Saunders, C. P. R. and Brooks, I. M.: The effects of high liquid water content on thunderstorm charging, J. Geophys. Res-Atmos., 97, 14 671–14 676, https://doi.org/10.1029/92JD01186, 1991.

Saunders, C. P. R. and Peck, S. L.: Laboratory studies of the influence of the rime accretion rate on charge transfer during crystal/graupel collisions, J. Geophys. Res. Atmos., 103, 13 949–13 956, https://doi.org/10.1029/97JD02644, 1998.

Saunders, C. P. R., Bax-Norman, H., Emersic, C., Avila, E. E., and Castellano, N. E.: Laboratory studies of the effect of cloud conditions on graupel/crystal charge transfer in thunderstorm electrification, Q. J. R. Meteorol. Soc., 132, 2653–2673, https://doi.org/10.1256/qj.05.218, 2006.

Seifert, A., Beheng, K.: A two-moment cloud microphysics parameterization for mixed-phase clouds. Part 1: Model description. Meteorol. Atmos. Phys., 92, 45–66. https://doi.org/10.1007/s00703-005-0112-4, 2006

Stolzenburg, M., W. D. Rust, and T. C. Marshall: Electrical structure in thunderstorm convective regions: 3. Synthesis, J. Geophys. Res., 103(D12), 14097–14108, https://doi.org/10.1029/97JD03545, 1988.

Straka, J. M. and Mansell, E. R.: A Bulk Microphysics Parameterization with Multiple Ice Precipitation Categories, J. Appl. Meteor., 44, 445– 466, https://doi.org/10.1175/JAM2211.1, 2005.

Sullivan, S. C., Hoose, C., Kiselev, A., Leisner, T., and Nenes, A.: Initiation of secondary ice production in clouds, Atmos. Chem. Phys, 18, 1593–1610, https://doi.org/10.5194/acp-18-1593-2018, 2018.

Sun, M., Liu, D., Qie, X., Mansell, E. R., Yair, Y., Fierro, A. O., Yuan, S., Chen, Z., and Wang, D.: Aerosol effects on electrification and lightning discharges in a multicell thunderstorm simulated by the WRF-ELEC model, Atmos. Chem. Phys, 21, 14 141–14 158, https://doi.org/10.5194/acp-21-14141-2021, 2021.

Takahashi, T.: Riming Electrification as a Charge Generation Mechanism in Thunderstorms, J. Atmos. Sci., 35, 1536–1548, https://doi.org/10.1175/1520-0469(1978)035<1536:REAACG>2.0.CO;2, 1978.

Taufour, M., Pinty, J.-P., Barthe, C., Vié, B., and Wang, C.: LIMA (v2.0): A full two-moment cloud microphysical scheme for the mesoscale non-hydrostatic model Meso-NH v5-6, Geosci. Model Dev., 17, 8773–8798, https://doi.org/10.5194/gmd-17-8773-2024, 2024.

Tsenova, B., Barthe, C., Mitzeva, R., and Pinty, J.-P.: Impact of parameterizations of ice particle charging based on rime accretion rate and effective water content on simulated with MésoNH thunderstorm charge distributions, Atmos. Res., 128, 85–97, https://doi.org/10.1016/j.atmosres.2013.03.011, 2013.

Vié, B., Pinty, J.-P., Berthet, S., and Leriche, M.: LIMA (v1.0): A quasi two-moment microphysical scheme driven by a multimodal population of cloud condensation and ice freezing nuclei, Geosci. Model Dev., 9, 567–586, https://doi.org/10.5194/gmd-9-567-2016, 2016.

---

## Referee Report (RR1)

The authors have addressed most of my comments. However, I still quite puzzled about the total lack of CWC to the SIP mechanism choices despite orders of magnitude changes in the ice crystal concentration. The authors provided some speculation about this, but none of their reasons would explain why there is no CWC sensitivity with SIP mechanism tests but substantial CWC sensitivity when the CCN concentration is varied. In Figure 6 (CCN tests), only the top middle panel shows a near lack of CWC sensitivity to CCN, but in this case the ice crystal number concentration is likewise not sensitive to CCN. The CWC analysis to me strongly suggests that there is an error in the data processing code. The authors state (Lines 454-55) that "As soon as one SIP process is activated, all mean vertical profiles of CWC are merged in the WARM and MID-WARM cases." What does this mean exactly? Is this merging the reason why the CWC profiles are so similar?

**Minor Comments:**

- 1. Line 88: Rather than "..." it would be better to use "etc." to end the sentence.
- 2. Line 125: "rimmed" → "rimed"

---

## Author Response (AR2)

**Reply on RC1**

**RC1:** The authors have addressed most of my comments. However, I still quite puzzled about the total lack of CWC to the SIP mechanism choices despite orders of magnitude changes in the ice crystal concentration. The authors provided some speculation about this, but none of their reasons would explain why there is no CWC sensitivity with SIP mechanism tests but substantial CWC sensitivity when the CCN concentration is varied. In Figure 6 (CCN tests), only the top middle panel shows a near lack of CWC sensitivity to CCN, but in this case the ice crystal number concentration is likewise not sensitive to CCN. The CWC analysis to me strongly suggests that there is an error in the data processing code. The authors state (Lines 454-55) that "As soon as one SIP process is activated, all mean vertical profiles of CWC are merged in the WARM and MID-WARM cases." What does this mean exactly? Is this merging the reason why the CWC profiles are so similar?

We understand the comment regarding the insensitivity of the cloud water content (CWC) to the secondary ice processes (SIP). The data processing scripts have been double-checked and no errors in the code have been found. Additionally, we have plotted the cloud droplet mixing ratio directly from the model outputs, and it shows the same tendencies as CWC. The same code is used to plot the ice crystal number concentrations (Figures 4 and 12) and it results in significantly different profiles for this variable for all sensitivity tests. Consequently, we are sure that this behavior cannot be attributed to an error in the processing code. Therefore, it must be related to the model itself, and in particular to the microphysics scheme, to the configuration of the sensitivity tests, and/or to the way the diagnostics is performed.

We first examined the sensitivity of CWC to the way the diagnostics is performed. In Figures 5 and 12, we plotted the mean vertical profiles of CWC in the convective region during the early cloud electrification. As stated in the manuscript "the initial stage of cloud electrification is defined as the first 10 min during which the absolute value of the non-inductive charging rate integrated over the volume of the convective region is greater than 0.1 C s-1". It corresponds approximately to the period around 20-25 min of simulation. We have investigated more thoroughly the sensitivity of CWC to  $N_{CCN}$ ,  $N_{INP}$  and SIP without the constraint of the electrification period defined in the manuscript.

Figure R1 shows the mean vertical profiles of cloud droplet mixing ratio in the convective region, for the 3 storms and at different times of the simulation, for sensitivity tests on SIP. At 22 min (first column of Fig. R1), quasi-identical vertical profiles of cloud droplet mixing ratio are found for each simulation and each idealized storm. However, at a later stage in the storm, small differences are found in all storms when SIP processes are activated, creating a sink of cloud droplets. For example, at 52 minutes in the COLD storm, differences of 0.05 g kg-1 are observed between the ALLSIP (0.25 g kg-1) and HM simulation (0.30 g kg-1) around 5 km altitude. This outcome arises from a complex balance between competing processes, notably autoconversion, riming and depositional growth.

The same figures as Figure R1 have been plotted for sensitivity tests on  $N_{\text{INP}}$  (Figure R2) and  $N_{\text{CCN}}$  (Figure R3). High  $N_{\text{INP}}$  tends to reduce cloud droplet mixing ratio in altitude in the WARM and MID-WARM storms at 22 minutes (Figure R2a and R2d) by accelerating the formation of ice crystals, and precipitation ice particles (aggregates and graupel) and

consuming cloud droplets. This effect is not observed in the COLD storm at 22 min because the development of the ice phase occurs sooner. Later in the storm life, the vertical profiles appear almost indistinguishable, except at 52 minutes in the WARM storm where high  $N_{\text{INP}}$  leads to higher cloud droplet mixing ratio.

Increasing  $N_{\text{CCN}}$  leads to a clear enhancement of cloud droplets mixing ratio during the whole lifecycle of the cloud, but with a higher impact at the development stage of the storms (22 min). Cloud droplet mixing ratio is reduced during the storm life mostly by riming resulting in similar vertical profiles regardless of  $N_{\text{CCN}}$  in the WARM and MID-WARM storms at 52 minutes (Figures R3c and R3f).

 $N_{CCN}$  directly impacts the formation and evolution of cloud droplets and it has naturally the highest impact on cloud droplet mixing ratio during the cloud development. On the contrary,  $N_{INP}$  and SIP processes have an indirect effect on cloud droplet mixing ratio and CWC through depositional growth and riming, and thus have a negligible impact during the early cloud electrification period compared to CCN. These figures also clearly show that the impact of  $N_{CCN}$ ,  $N_{INP}$  and SIP on CWC depends on the period of the storm during which the diagnostics are computed.

We have also tested the impact of the region over which the diagnostics are made. In the manuscript, the mean vertical profiles are computed in the convective region where the cloud electrification is supposed to mainly occur. We have computed the mean vertical profiles of CWC during the early electrification period in the stratiform region and in the whole cloud for the sensitivity tests on SIP processes (not shown). We clearly observe different CWC profiles between the NOSIP simulation and the simulations where at least one SIP process is activated in the WARM and MID-WARM storms. The cloud region over which the average is calculated can also influence the effect of  $N_{\text{CCN}}$ ,  $N_{\text{INP}}$  and SIP on CWC.

Part of the behavior of CWC could also be attributed to the configuration of sensitivity tests on SIP processes. Indeed, all sensitivity tests on SIP have been performed with fixed  $N_{\rm CCN}$  (1000 cm-3) and  $N_{\rm INP}$  (10 L-1). We performed additional simulations to investigate whether changing these values would modify the behavior of CWC profiles.  $N_{\rm CCN}$  and  $N_{\rm INP}$  have been set to 5000 cm-3 and 100 L-1, and two simulations with none SIP and all SIP activated have been performed for the WARM case. Figure R4 shows the mean vertical profiles of cloud droplet mixing ratio for these simulations. There is almost no impact of the SIP processes on the cloud droplet mean vertical profile for the WARM case. The impact of SIP processes when  $N_{\rm INP}$  and  $N_{\rm CCN}$  are increased is even less pronounced at 52 min than in the original set of experiments (1st line of Figure R1). We also tested lower aerosol concentrations:  $N_{\rm CCN}$  and  $N_{\rm INP}$  have been set to 500 cm-3 and 10 L-1, respectively. Results are shown in Figure R5. As in other cases, differences in the mean vertical profiles of cloud droplet mixing ratio widen over time. But as soon as 37 minutes, slight differences are observed. The maximum difference is reached at 52 min between the NOSIP and ALLSIP simulations (0.02 g kg-1). The sensitivity of CWC to SIP processes seems to increase at lower  $N_{\rm CCN}$ .

Finally, and as stated in the first review, some microphysical treatment in LIMA v1.0 (Vié et al., 2016) could affect the insensitivity of CWC to ice particles. Four additional simulations of the WARM storm have been performed using the full two-moment version of the LIMA microphysics scheme (LIMA v2.0; Taufour et al., 2024). In all these simulations,  $N_{\text{CCN}}$  and  $N_{\text{INP}}$  are set to 1000 cm-3 and 10 L-1, respectively. The SIP processes are enabled or disabled in turn. Figure R6 shows the mean vertical profiles of the cloud droplet mixing ratio at 22, 37

and 52 min for the four sensitivity tests using LIMA v2.0. At 22 min, there is almost no impact of activating or not the SIP processes. However, at 37 and 52 min, the impact of the SIP processes is much more important when using LIMA v2.0 than when using LIMA v1.0. As stated in the manuscript, "snow and graupel number concentrations are not prognostic, potentially accelerating their formation and depleting liquid and small ice species in comparison with a full two-moment version of LIMA". In LIMA v2.0 the formation of snow/aggregates and graupel is more gradual.

To summarize, additional simulations enable to conclude that:

- Increasing the initial  $N_{CCN}$  and  $N_{INP}$  does not change the effect of SIP processes on the CWC vertical profile during the early cloud electrification period. It even diminishes the long-term effect. But, reducing the initial  $N_{CCN}$  enhances the effect of SIP processes on CWC, though this remains limited to the mature stage of the storm.
- Using the full two-moment version of the LIMA scheme does not change the CWC results during the early electrification period. However, a contrasted behavior of CWC when SIP processes are activated is observed as soon as the storm reaches its mature stage.

Regarding the use of "merge" in lines 454-55, it is done to describe the visual appearance of the curves rather than a technical process. This term has been changed to "almost indistinguishable" to avoid confusion in the new version of the manuscript.

Therefore the paragraph discussing the insensitivity of CWC to SIP processes in Section 4.2.4 has been modified to better explain the potential origins of such a behavior.

"The weak sensitivity of CWC to SIP processes may result from several factors: data sampling, configuration of the sensitivity tests, and configuration of the microphysics scheme. First, data sampling during the electrification period limits the detection of differences which occur more significantly during the storm's mature stage (not shown). While  $N_{CCN}$  directly impacts the formation of cloud droplets,  $N_{INP}$  and SIP processes have a more indirect effect on cloud droplets. They impact the cloud water content through depositional growth and riming. Thus their effect is delayed in time compared to  $N_{\text{CCN}}$  and is negligible during the early cloud electrification period investigated in this study. Secondly, additional simulations for the WARM case have been performed in which  $N_{CCN}$  and  $N_{INP}$  were varied, and the SIP processes were enabled or disabled. Increasing the initial  $N_{CCN}$  and  $N_{INP}$ does not change the effect of SIP processes on the CWC vertical profile during the early cloud electrification period. It even diminishes the long-term effect. But, reducing the initial NCCN enhances the effect of SIP processes on CWC, though this remains limited to the mature stage of the storm. Finally, in the version of LIMA used in this study, snow and graupel number concentrations are not prognostic, potentially accelerating their formation and depleting liquid and small ice species in comparison with a full two-moment version of LIMA (Taufour et al., 2024). Using the full two-moment version of the LIMA scheme in the WARM simulation and enabling or disabling the SIP processes does not change the CWC results during the early electrification period. However, a contrasted behavior of CWC when SIP processes are activated is observed as soon as the storm reaches its mature stage."

**Minor Comments:**

1. Line 88: Rather than "..." it would be better to use "etc." to end the sentence.

Done.

2. Line 125: "rimmed" → "rimed"

Done.

Figure R1: Effect of time and SIP processes. Mean vertical profiles of cloud droplet mixing ratio (g kg-1) in the convective region at 22 min (left column), 37 min (middle column) and 52 min (right column) of the WARM (first line), MID-WARM (second line), and COLD (third line) simulations. In each panel, the blue, black, green, orange and pink curves correspond to the mean vertical profiles of cloud droplet mixing ratio for the NOSIP, HM, HM+CIBU, HM+RDSF and ALLSIP simulations, respectively. In all these simulations, NCCN and NINP are set to 1000 cm-3 and 10 L-1, respectively.

Figure R2: Effect of time and  $N_{\text{INP}}$ . Mean vertical profiles of cloud droplet mixing ratio (g kg-1) in the convective region at 22 min (left column), 37 min (middle column) and 52 min (right column) of the WARM (first line), MID-WARM (second line), and COLD (third line) simulations. In each panel, the green, orange and pink curves correspond to the mean vertical profiles of cloud droplet mixing ratio for  $N_{\text{INP}} = 10$ , 100 and 1000 L-1, respectively. In all these simulations,  $N_{\text{CCN}}$  is set to 1000 cm-3 and only the HM process is activated.

Figure R3: Effect of time and  $N_{CCN}$ . Mean vertical profiles of cloud droplet mixing ratio (g kg-1) in the convective region at 22 min (left column), 37 min (middle column) and 52 min (right column) of the WARM (first line), MID-WARM (second line), and COLD (third line) simulations. In each panel, the blue, black, green, orange and pink curves correspond to the mean vertical profiles of cloud droplet mixing ratio for  $N_{CCN}$  = 500, 1000, 5000, 8000 and 10000 cm-3, respectively. In all these simulations,  $N_{INP}$  is set to 10 L-1 and only the HM process is activated.

Figure R4: Effect of the initial aerosol concentration. Mean vertical profiles of cloud droplet mixing ratio (g  $kg^{-1}$ ) in the convective region at 22 min (a), 37 min (b) and 52 min (c) of the WARM simulations. In each panel, the blue, black and pink curves correspond to the mean vertical profiles of cloud droplet mixing ratio for the NOSIP, HM and ALLSIP sensitivity tests, respectively. In all these simulations,  $N_{CCN}$  and  $N_{INP}$  are set to 5000 cm-3 and 100 L-1, respectively.

Figure R5: Effect of the initial aerosol concentration. Mean vertical profiles of cloud droplet mixing ratio (g  $kg^{-1}$ ) in the convective region at 22 min (a), 37 min (b) and 52 min (c) of the WARM simulations. In each panel, the blue, black and pink curves correspond to the mean vertical profiles of cloud droplet mixing ratio for the NOSIP, HM and ALLSIP sensitivity tests, respectively. In all these simulations,  $N_{CCN}$  and  $N_{INP}$  are set to 500 cm-3 and 10 L-1, respectively.

Figure R6: Effect of the microphysics scheme. Mean vertical profiles of cloud droplet mixing ratio (g kg $^{-1}$ ) in the convective region at 22 min (a), 37 min (b) and 52 min (c) of the WARM simulations. In each panel, the blue, black, green and pink curves correspond to the mean vertical profiles of cloud droplet mixing ratio for the NOSIP, HM, CIBU and ALLSIP sensitivity tests, respectively. In all these simulations,  $N_{CCN}$  and  $N_{INP}$  are set to 1000 cm $^{-3}$  and 10 L $^{-1}$ , respectively, and the full two moment scheme described in Taufour et al. (2024) is used.

---

## Author Response (AR3)

**Reply to Editor**

The authors have convincingly explained the origin of the lack of CWC sensitivity to SIP processes. For clarity of the presentation, it seems advised to clarify the contribution of data sampling to the insensitivity and the effect of averaging over the initial stage of electrification.

We thank the editor for her time and efforts in reviewing our manuscript.

We added a clarification on data sampling in the convective region in the manuscript:

"Data sampling in the convective region also contributes to the apparent insensitivity of CWC to SIP processes. Testing different vertical velocity and precipitation thresholds to define the convective zone did not change this result. However, the vertical profiles in the stratiform region reveal more pronounced differences."